# AdaSplats: Adaptive Splatting of Point Clouds for Accurate 3D Modeling and Real-Time High-Fidelity LiDAR Simulation

Jean Pierre Richa [1,2,*], Jean-Emmanuel Deschaud [1], François Goulette [1,3] and Nicolas Dalmasso [2]

1    Centre for Robotics, Mines Paris, PSL University, 75006 Paris, France
2    ANSYS France, 15 Pl. Georges Pompidou, 78180 Montigny-le-Bretonneux, France
3    U2IS, ENSTA Paris, Institut Polytechnique de Paris, 91120 Palaiseau, France
*    Correspondence: jean-pierre.richa@minesparis.psl.eu or jeanpierre.richa@ansys.com

**Abstract:** LiDAR sensors provide rich 3D information about their surroundings and are becoming increasingly important for autonomous vehicles tasks such as localization, semantic segmentation, object detection, and tracking. Simulation accelerates the testing, validation, and deployment of autonomous vehicles while also reducing cost and eliminating the risks of testing in real-world scenarios. We address the problem of high-fidelity LiDAR simulation and present a pipeline that leverages real-world point clouds acquired by mobile mapping systems. Point-based geometry representations, more specifically splats (2D oriented disks with normals), have proven their ability to accurately model the underlying surface in large point clouds, mainly with uniform density. We introduce an adaptive splat generation method that accurately models the underlying 3D geometry to handle real-world point clouds with variable densities, especially for thin structures. Moreover, we introduce a fast LiDAR sensor simulator, working in the splatted model, that leverages the GPU parallel architecture with an acceleration structure while focusing on efficiently handling large point clouds. We test our LiDAR simulation in real-world conditions, showing qualitative and quantitative results compared to basic splatting and meshing techniques, demonstrating the interest of our modeling technique.

**Keywords:** point clouds; 3D modeling; splatting; surface reconstruction; LiDAR simulation; ray tracing

## 1. Introduction

Constructing virtual environments inside which autonomous vehicles (AVs) and their sensors can be simulated is not an easy task. This can be achieved by 3D artists who carefully craft the scenes manually, such as in CARLA simulator [1]. Although handcrafted simulators provide leverage for testing AV algorithms, they introduce a large domain gap with respect to real-world environments. This gap arises from the difference between the almost perfect geometry present in such simulators, as they contain carefully designed simple 3D objects, which results in simplistic environments. Improving realism, manual construction involves months of work and associated costs (e.g., creating photo-realistic scenes costs 10,000 dollars per kilometer [2]).

The limitations of manually constructed simulation environments have given rise to simulators [2,3] using real-world point clouds collected from a LiDAR scanner mounted on a mobile mapping system (MMS). These methods model the real-world automatically from outdoor point clouds using well-known splatting techniques [4,5]. Splats are oriented 2D disks that approximate the local neighborhood by following the curvature of the surface and are defined by a center point, a normal vector, and a radius. They are used as geometric primitives to model the 3D surface. These simulation methods follow early work carried out on point-based modeling and rendering [6] to achieve high-quality 3D modeling while reducing the number of generated geometric primitives. They have been proposed for simulating the LiDAR sensor in the splatted environment, resulting in higher accuracy on

tasks such as semantic segmentation (SS) and vehicle detection learned from their simulated data, when compared with data collected from handcrafted simulators. However, these methods neither demonstrate accurate geometric modeling of reality from MMS point clouds, nor do they address the time aspect in LiDAR simulation.

A classical approach to automatic 3D modeling is surface reconstruction, which can be performed on point clouds acquired using LiDAR sensors. Surface reconstruction is a well studied area of research, and many algorithms have been proposed to reconstruct an explicit surface representation from unorganized point sets in the form of triangular meshes [7–9]. However, most existing methods fail to represent complex structures, especially open shapes and thin structures, in a real outdoor environment.

Some approaches have proposed generating a hybrid mesh-splat surface representation [10,11]. However, they either do not focus on accurate geometry representation, or in the case of the latter [11], use very expensive mesh generation techniques.

Accurate geometry representation of large outdoor point clouds is essential for high-quality sensor simulation. In this work, we introduce adaptive splats (AdaSplats) to overcome the limitations of previous splatting and meshing techniques by using local geometric cues and point-wise semantic labels. Splat modeling is more flexible than a mesh reconstruction, and the size of each splat can be adapted by semantic or local geometric information.

SS on 3D point clouds is the task of assigning point-wise semantic labels. There exist a plethora of methods for the SS task [12–15]. We leverage KPConv [12] for its reported accuracy on a variety of datasets [16–18]. For example, KPConv achieves an average mean Intersection over Union (mIoU) of 82% on the Paris-Lille-3D dataset [16]. This is an outdoor dataset collected using a LiDAR mounted on a MMS and close to the Paris-CARLA-3D (PC3D) dataset [19] that we use in our experiments. Using KPConv, we perform SS on the outdoor point clouds to obtain point-wise semantic labels that are later used for the generation of AdaSplats.

LiDAR point clouds acquired using a MMS are highly anisotropic [16]. Resampling the point cloud reduces the anisotropy and increases uniformity by redistributing the points. Although previous upsampling methods [20,21] have addressed this problem, they require passing through computationally expensive representations. Deep learning methods for point cloud upsampling [22,23] are used to increase the uniformity of the points distribution, and deep learning for depth completion [24,25] can also help in the case of sparse point clouds. However, these methods are data hungry and usually limited to small objects or scenes. We propose a point cloud resampling that exploits our splats modeling, eliminating the need for extensive data preprocessing and training while achieving isotropic resampling.

Physics-based sensor simulation, such as simulation of the camera and LiDAR sensors, requires ray tracing the generated primitives. Previous splat ray-tracing algorithms [26] are limited to ray–splat intersection using a CPU parallelized implementation. Accelerating sensor simulation requires achieving real-time ray tracing, which is not possible to achieve with CPU parallelization. We introduce a ray–splat intersection method leveraging the GPU architecture and an acceleration data structure, achieving real-time rendering in the splat model.

We propose a fully automatic pipeline for accurate 3D modeling of outdoor environments and simulation of a LiDAR sensor from real-world data (see Figure 1).

More specifically, and different from previous LiDAR simulators leveraging existing point cloud splatting techniques [2,3], we focus on accurate geometry representation of point clouds collected in outdoor environments and introduce AdaSplats. We propose the following two variants of AdaSplats: The first uses point-wise semantic labels obtained from the predictions of a deep network; and the second uses only local neighborhood descriptors obtained from analysis on the principal components describing the local surface. The second variant is introduced for use in the absence of large ground truth semantic information to train neural networks.

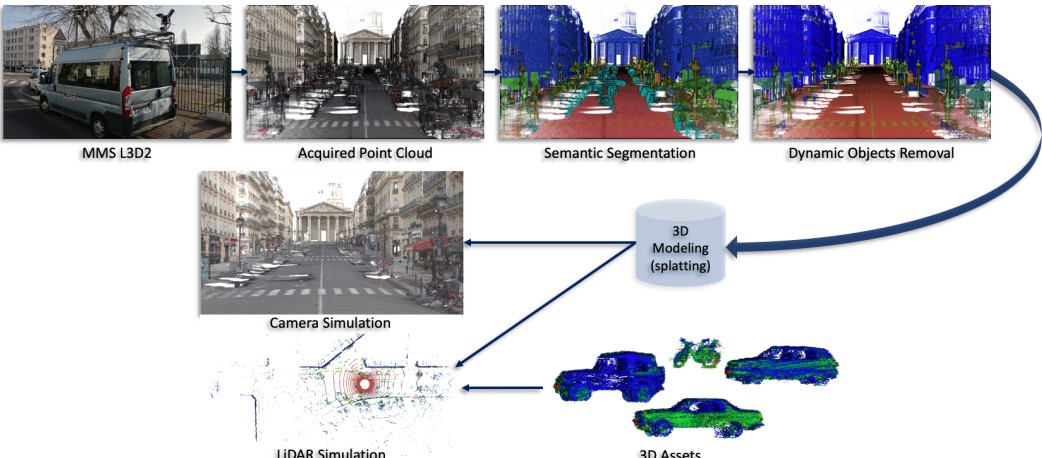

**Figure 1.** Starting with a point cloud acquired using a mobile mapping system (MMS), we obtain point-wise semantic labels by performing semantic segmentation. Using the semantic labels, we remove dynamic objects in the scene and perform our splats generation method. The splatted scene can then be used to simulate the different sensors. Dynamic objects can be added to the splatted scene either in the form of splatted point clouds or using a bank of CAD meshed models.

The proposed pipeline aims to generate an accurate 3D model of outdoor scenes represented by point clouds collected using a range sensor (mobile or static), which has not previously been discussed in the context of sensor simulation [2,3]. Moreover, it targets speeding up the simulation by reducing the number of generated primitives and accelerating the ray–primitive intersection. Our pipeline is highly dynamic and can be modified to include intermediate steps, such as the addition of dynamic objects (e.g., moving vehicles and pedestrians). This is demonstrated in the experiments in Section 6.3.4, where we introduce the SimKITTI32 dataset, a simulated Velodyne HDL-32 LiDAR in a splatted environment from SemanticKITTI acquired by a Velodyne HDL-64 [18].

Our contributions can be summarized as follows:

- AdaSplats: a novel adaptive splatting approach for accurate 3D geometry modeling of large outdoor noisy point clouds;
- Splat-based point cloud resampling, dealing with highly varying densities and scalable to large data;
- Faster-than-real-time GPU ray casting in the splat model for LiDAR sensor simulation and rendering;
- SimKITTI32: a dataset simulating a Velodyne HDL-32 inside a sequence of SemanticKITTI dataset [18]. It is publicly available at: https://npm3d.fr/simkitti32 (accessed on 5 December 2022).

## 2. Related Works

We introduce surface reconstruction methods and state their limitations; then, we talk about point-based modeling and resampling techniques that allow us to achieve a more realistic surface representation with a lower number of geometric primitives. Finally, we introduce previous LiDAR simulation methods performed in splatted scene models.

### 2.1. Surface Reconstruction

The earliest methods on surface representation from point clouds mainly focused on surface reconstruction [7] producing triangular meshes. These methods were designed to work on closed surfaces and required a good point-wise normals estimation with no errors in orientation, which is a complex problem and an active area of research [27–30]. Using distance functions on 3D volumetric grids for surface reconstruction traces back to

the seminal work of Hoppe et al. [7], in which a signed distance field $\phi : \mathbb{R}^3 \to \mathbb{R}$ was used to represent the underlying surface from a point cloud, as follows:

$$\phi(x) = \hat{\mathbf{n}} \cdot (x - \mathbf{p}) \tag{1}$$

where $x$ is the voxel coordinates in the 3D grid, $\mathbf{p}$ is the nearest neighbor of $x$ in the point cloud $\mathcal{P}$, and the vector $\mathbf{n}$ with a hat ($\hat{\mathbf{n}}$) is the normal unit vector associated with the point $\mathbf{p}$.

Having computed the signed distance function (SDF) on a regular 3D grid, the marching cubes algorithm [31] extracts the final iso-surface as a mesh.

Following this pioneering work, several methods have been proposed to deal with noisy data, such as Implicit Moving Least Squares (IMLS) [8], which approximates the local neighborhood of a given voxel in the grid as a weighted average of the local point functions, as follows:

$$\mathrm{IMLS}(x) = \frac{\sum_{\mathbf{p}_k \in \mathcal{N}_x} \hat{\mathbf{n}}_k \cdot (x - \mathbf{p}_k) \, \theta_k(x)}{\sum_{\mathbf{p}_k \in \mathcal{N}_x} \theta_k(x)} \tag{2}$$

where $x$ is the voxel coordinates, $\mathcal{N}_x$ is the set of $\mathbf{p}_k$ neighboring points from $x$ in the point cloud $\mathcal{P}$, $\hat{\mathbf{n}}_k$ is the normal unit vector associated with the point $\mathbf{p}_k$, and $\theta_k$ is the Gaussian weight defined as:

$$\theta_k(x) = e^{-||x - \mathbf{p}_k||_2^2 / \sigma^2} \tag{3}$$

where $\sigma$ is a parameter of the influence of points in $\mathcal{N}_x$.

Other surface reconstruction methods [9,32] use global implicit functions, such as indicator functions where the reconstruction problem is solved using a Poisson system equation.

### 2.1.1. Volumetric Segmentation

Volumetric segmentation is a subcategory of indicator function that classifies whether a voxel is occupied or empty with a confidence level using octrees [33] or Delaunay triangulation [34]. It can be scaled to arbitrarily large point clouds by distributing the surface reconstruction problem [35]. Some works on surface reconstruction have focused on identifying drive-able zones for robot navigation through the creation of simplistic 3D models of roads and buildings [36]. Others have proposed improvements for detailed facade reconstruction [37]. Some applications have real-time constraints, like in [38]. However, they do not create watertight meshes, and they introduce many disconnected parts and holes. More recently, 3DConvNets [39,40] are applied for surface reconstruction but need large training data.

### 2.1.2. Volumetric Fusion

In a different approach, the volumetric fusion method of VRIP [41] takes advantage of range images. It creates a mesh from the depth image to cast a ray from the sensor origin to the voxel of the volumetric grid, obtaining a signed distance to the mesh. Then, it merges the scans' distances in a least-squares sense. However, this distance field can only be computed from range images and cannot be used directly on point clouds.

### 2.2. Point-Based Surface Modeling

Point-based rendering gained interest after the report published by Levoy and Whitted on using points as display primitives [6].

### 2.2.1. Splatting

The first methods on rendering such primitives without any connectivity information [5,42] focused on achieving a low rendering time and interactively displaying large amounts of points. They used accelerating hierarchical data structures to accelerate the rendering of the generated splats, which are oriented 2D disks expanding to generate a

hole-free approximation of the surface. They used splats as their modeling and rendering primitives.

High-Quality Rendering

Using splats as rendering primitives became the best choice for surface modeling after their efficiency and effectiveness were proven in many methods. Surface Splatting [4] introduces a point rendering and texture filtering technique and achieves high quality anisotropic anti-aliasing. It combines oriented 2D reconstruction kernels, circular, or elliptical splats, with a band-limiting image-space elliptical weighted average (EWA) texture filter. Other methods in this area [43,44] exploit the programmability of modern GPUs and detail the best practices for rendering point-based methods using elliptical splats. They achieve a high frame rate even in the presence of a high number of primitives, which makes it possible for real-time applications to contain more details than similar scenes based on polygonal meshes. Surfels [5] is another approach that focuses on the accurate mapping of textures to splats to increase the visual details of the rendered objects while rendering at interactive rates.

Achieving a high frame rate in the presence of a high number of primitives is essential. However, it is also important to reduce the preprocessing computational complexity as much as possible. One approach using circular or elliptical splats [45] creates a hole-free approximation of the surface and then performs a relaxation procedure that results in a minimal set of splats that best estimates the surface.

Advanced Shading

The aforementioned approaches do not focus on generating photo-realistic rendering because they use one normal vector per splat, which results in rendering that is comparable with Gouraud or flat shading. A previous method overcomes this challenge by associating a normal field to each splat that is created using the normals of points in the neighborhood of the splat center [46]. This approach provides a better local approximation of the surface normals and results in rendering comparable to Phong shading for regular meshes, hence the name Phong splatting. However, it is computationally expensive to generate a per-splat normal field. Moreover, it increases the host (CPU) to device (GPU) communication time and memory footprint resulting from transferring more information to the device.

2.2.2. Splats Ray Tracing

The previous approaches focused on several aspects of using points as rendering primitives. However, one important aspect that is still missing is ray tracing the generated splats. In [26], the authors leverage the previous methods while modifying the pipeline to best suit their ray tracing algorithm. They generate a hole-free approximation of the surface. Moreover, they associate each splat with a normal field and then ray trace the generated splats, performing per-pixel Phong shading. They achieve photo-realistic rendering on dense and uniformly distributed point clouds while using an octree to accelerate the ray–splat intersection.

Leveraging this last method, which was implemented to work only on CPUs, we implement an efficient GPU ray–splat intersection, improve their splats generation method, and use semantic information obtained from deep neural networks to build adaptive splats.

*2.3. Neural Radiance Fields*

More recent methods, such as neural radiance fields (NeRF) [47], have gained a lot of interest in the rendering and novel view synthesis communities. They train neural networks to generate photo-realistic novel views of a scene from a set of calibrated input images. Several subsequent methods [48–52] have introduced different approaches to reduce the computational complexity and providing real-time inference ability. These methods achieve impressive rendering quality of complex scenes. However, we focus our

work on high-fidelity LiDAR simulation and leave photo-realistic camera simulation for future works.

*2.4. Resampling*

Data sparsity and nonuniformity dominate point clouds collected using a MMS, as shown in [16]. Isotropic resampling to increase uniformity of the points distribution across the point cloud facilitates the splats generation and improves the normals estimation. A previous upsampling approach [20] works directly on the geometry inferred from the local neighborhood of points in the point cloud. The method begins by computing an approximation of the Voronoi diagram of neighborhood points at a random point. Then, it chooses the Voronoi vertex whose circle has the largest radius and projects the vertex on the surface with the moving least squares (MLS) projection. The process is repeated until the radius of the largest circle is less than a defined threshold. This results in an upsampling that accurately approximates the surface geometry. However, Voronoi diagram approximations are expensive to compute.

Mesh-based resampling techniques can be achieved by reconstructing the surface from the acquired point cloud, simplifying the mesh while preserving the local underlying surface structure [53], and then sampling points on the reconstructed surface. Although they can achieve a good approximation of the surface, they are dependent on geometry errors introduced by surface reconstruction methods.

Other methods use deep learning techniques directly on point clouds to achieve higher density on sparse point clouds through upsampling [22,23]. Depth completion can also be used to infer the completed depth map from an incomplete one, which can later be re-projected into 3D to upsample the point cloud [24,25]. However, current deep learning methods are limited to small scenes and suffer from a performance drop with unseen real-world data.

Inspired by [20], we introduce a novel splat-based point cloud resampling approach that increases the uniformity of points distribution. Moreover, we embed a denoising and an outlier rejection step into the sampling algorithm that helps with achieving a minimal set of points that will be used later to generate the splats. With the resampled point cloud, we achieve high-quality, hole-free surface modeling using our adaptive splats approach.

*2.5. LiDAR Simulation*

The availability of handcrafted simulated environments such as CARLA [1], BlenSor [54], and game engines (GTA-V) offers the ability to simulate LiDAR sensors and collect scans [55–57]. Deep learning methods leverage the huge amount of data that can be collected from such environments. However, they introduce a large domain gap between synthetic and real-world data. Although this gap can be reduced with domain adaptation strategies [58,59], it still limits the ability of trained neural networks to generalize to the real world when trained on synthetic simulated LiDAR data.

2.5.1. Volumetric Scene Representation

A previous approach [60], extended in [61], focuses on accurate interaction between the LiDAR beam and the environment. In this work, the environment was modeled from real LiDAR data to reduce the domain gap. The authors introduce permeability to sample the points of intersection from 3D Gaussian kernels contained in volumetric grids. Although good results can be achieved, a volumetric representation is not accurate for modeling the underlying surface, and the approach is computationally expensive, so it cannot be used for real-time applications.

2.5.2. Splat-Based Scene Representation

More recent approaches [2,3] acquire data using a LiDAR mounted on a MMS and model the 3D geometry using splatting techniques after removing the dynamic objects in the foreground (e.g., pedestrians, cars, etc.). These approaches add dynamic objects on top

of the reconstructed 3D environment in the form of CAD models, such as in [2], or in the form of point clouds collected from the real world, such as in [3]. In the first approach [2], the authors do not take into account the physical model of the LiDAR, since they do not cast rays; instead, they use cube maps rasterization to accelerate the simulation. In the second approach [3], they use Embree [62] to accelerate the ray–primitive intersection. However, Embree runs on CPU and is still far from real-time since the parallelization of the ray casting is limited to the number of CPU cores. They achieve higher performance on object detection and SS tasks using data collected from their simulator, compared to data collected from CARLA. However, they do not focus on demonstrating accurate modeling of the static background, which prove to play the most important part in elevating the deep neural networks' performance in their experiments.

### 2.5.3. Mesh-Based Scene Representation

A more recent work [63] simulates an aerial laser scanner to scan a reconstructed scene as a base model. After scanning, the authors use distance metrics to evaluate the different reconstruction algorithms. However, they use a meshed model as ground truth, which is subject to reconstruction error. Instead, we compare the simulated point cloud to raw data available as point cloud.

### 2.5.4. Real-Time LiDAR Simulation

LiDAR simulation can be done offline. However, achieving real-time simulation is important for accelerating AV testing and validation. Ray casting is used for physics-based LiDAR simulation by casting rays from the virtual sensor placed in the virtual scene. Ray casting is highly parallelizable and can be further accelerated through the use of accelerating structures. Embree [62] is a ray-tracing engine working on CPU that builds an acceleration structure to reduce the computation time by accelerating the ray–primitive intersection. Although it drastically reduces the ray-tracing time, it is still limited to the use of CPU cores to parallelize the ray casting. OptiX [64], on the other hand, builds a bounding volume hierarchy (BVH) structure to arrange the geometric primitives in a tree and benefits from the parallel architecture of GPUs to further accelerate the ray-casting. This reduces the time even further, compared with Embree. To this end, we choose to use OptiX and implement the ray–splat intersection using CUDA to accelerate the intersection process and achieve real-time LiDAR simulation.

## 3. Adaptive Splatting

In this section, we describe our AdaSplats approach for generating a high-quality and hole-free approximation of the underlying surface from a point cloud. We begin by introducing the adaptive splats generation algorithm, followed by our resampling method. In our approach, we leverage the accuracy of the state-of-the-art deep learning method KPConv [12] in SS of 3D outdoor point clouds.

Using the semantic information, we adapt the splat growing and generation to better model the geometry. We also use the semantic information in the resampling process.

### 3.1. Basic Splatting

We describe here a variant of the splatting method of Linsen et al. [26], the basic algorithm on which we develop our adaptive method.

We consider as input data a point cloud $\mathcal{P} = \{\mathbf{p}_i \in \mathbb{R}^3 \mid 0 \leq i \leq \mathcal{N}\}$. We first compute an average radius $\bar{\mathcal{R}}$ of points in the $\mathcal{K}$-nearest neighbors neighborhood of every point, $\mathbf{p}_i$. For all experiments, we choose $\mathcal{K} = 40$. We then define $\mathcal{N}_{\mathbf{p}_i}$, the neighborhood of $\mathbf{p}_i$ to be the smallest neighborhood between $\mathcal{K}$-nn and a sphere of radius $\bar{\mathcal{R}}$, to be robust to the highly variable density of points. We perform (PCA) on $\mathcal{N}_{\mathbf{p}_i}$ to obtain the normal $\hat{\mathbf{n}}_i$ at each point $\mathbf{p}_i$ and reorient it with respect to the LiDAR sensor position.

Each splat $S_i$ is defined by $(\mathbf{c}_{S_i}, \hat{\mathbf{n}}_{S_i}, r_{S_i})$, with $\mathbf{c}_{S_i}$ being the center of the splat, $\hat{\mathbf{n}}_{S_i}$ the unit normal vector, and $r_{S_i}$ the radius of the splat. A splat $S_i$ centered at $\mathbf{p}_i \in \mathcal{P}$ initially has

$\hat{\mathbf{n}}_{S_i} = \hat{\mathbf{n}}_i$ and $r_i = 0$, which is increased by including points in $\mathcal{N}_{\mathbf{p}_i}$ (sorted in the order of increasing distance to $\mathbf{p}_i$). We compute the signed point-to-plane distance of each neighbor $\mathbf{p}_i^k \in \mathcal{N}_{\mathbf{p}_i}$:

$$\epsilon_i^k = \hat{\mathbf{n}}_i \cdot (\mathbf{p}_i^k - \mathbf{p}_i) \tag{4}$$

We stop the growing in $\mathcal{N}_{\mathbf{p}_i}$ when $|\epsilon_i^k|$ exceeds an error bound $\bar{\mathcal{E}}$ (see below). When the growing is done, we update the splat's center position by moving it along the normal:

$$\mathbf{c}_{S_i} = \mathbf{p}_i + \bar{\epsilon}_i \, \hat{\mathbf{n}}_i \tag{5}$$

with $\bar{\epsilon}_i$ being the average signed point-to-plane distance of the points included in its generation. We then set the radius of the splat as the projected distance of the farthest point $\mathbf{p}_i^{k_{last}}$, which is the last neighbor having the point-to-plane distance below the $\bar{\mathcal{E}}$ threshold, from $\mathbf{c}_{S_i}$:

$$r_{S_i} = ||(\mathbf{p}_i^{k_{last}} - \mathbf{c}_{S_i}) - \hat{\mathbf{n}}_i \cdot (\mathbf{p}_i^{k_{last}} - \mathbf{c}_{S_i}) \, \hat{\mathbf{n}}_i||_2 \tag{6}$$

Then, all points in the neighborhood $\mathcal{N}_{\mathbf{p}_i}$ inside the sphere of radius $\alpha \, r_{S_i}$ are discarded from the splat generation. $\alpha$ is a global parameter in $[0, 1]$ that allows the entire surface to be covered without holes while minimizing the number of splats generated (we used $\alpha = 0.2$ for all experiments).

Before starting the generation process, we compute the error bound $\bar{\mathcal{E}}$ as the average unsigned point-to-plane distance of points in all $\mathcal{N}_{\mathbf{p}_i}$. Finally, we keep the $m$ splats with radius $r_{S_i} > 0$, where $m$ is much lower than the number of points $\mathcal{N}$ in the point cloud. Figure 2 provides a visual illustration of the splats generation process.

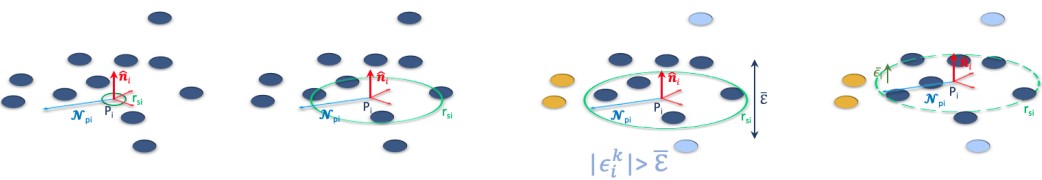

**Figure 2.** Splats generation starts by including points in the neighborhood, until the error bounds are exceeded, then the center of the splat is moved along the normal vector to minimize the distance from the splat to the neighboring points.

Now that we have introduced the method that we call Basic Splats generation (which we use as a baseline), we move on to explain our adaptive splats (AdaSplats) generation using semantic information. AdaSplats uses the steps introduced in the basic splats generation and adds the point-wise semantic labels to improve the modeling quality.

### 3.2. Adaptive Splatting

We first perform SS on the raw point cloud using deep learning [12] to obtain the semantic classes in the point cloud. We then remove the detected points classified as dynamic objects (moving and parked cars, moving pedestrians, cyclists, etc.). Using the semantic information, we divide the points into the following four main groups:

- Ground: road and sidewalk;
- Surface: buildings and other similar classes that locally resemble a surface;
- Linear: poles, traffic signs, and similar objects;
- Non-surface: vegetation, fences, and similar objects.

In the adaptive splats generation, based on the group of the starting point $\mathbf{p}_i$, we change the neighborhood $\mathcal{N}_{\mathbf{p}_i}$ with parameters $\mathcal{K}$ and $\bar{\mathcal{R}}$ (as a reminder, $\mathcal{N}_{\mathbf{p}_i}$ is the smallest neighborhood between $\mathcal{K}$-nn and a sphere of radius $\bar{\mathcal{R}}$). Moreover, we change the error bound parameter $\bar{\mathcal{E}}$, the criterion that stops the growth of the splats. The parameters used for the four groups are as follows:

- Ground: $3\mathcal{K} = 120$, $3\bar{\mathcal{R}}$, $3\bar{\mathcal{E}}$;
- Surface: $\mathcal{K} = 40$, $\bar{\mathcal{R}}$, $\bar{\mathcal{E}}$ (no change compared to basic splat);
- Linear: $0.33\mathcal{K} = 13$, $0.33\bar{\mathcal{R}}$, $0.33\bar{\mathcal{E}}$;
- Non-surface: $0.25\mathcal{K} = 10$, $0.25\bar{\mathcal{R}}$, $0.25\bar{\mathcal{E}}$.

The choice of parameters follows the analysis on points density to identify one that is sufficient to locally resemble a plane on the different class groups. The analysis takes into consideration the maximum size of a generated splat on a given structure that does not result in a geometry deformation. For example, a big splat on a linear structure, or a tree leaf, and do not leave holes in large planar areas like the ground. Through visual inspection, we observe the impact of varying the splats parameters for the different class groups. Finally, the parameters were chosen as the best set that could be used on the three datasets tested in the experiments section.

We also stop growing splat $S_i$ when a new point $\mathbf{p}_i^k \in \mathcal{N}_{\mathbf{p}_i}$ has a semantic class different from the class of $\mathbf{p}_i$.

These two adaptations in the growing of splats help to better model the geometry depending on the group and the semantics of points (e.g., improving splats for fine structures or the vegetation); they also improve the geometry at the intersection of different semantic areas and provide the ability to recover larger missing regions in ground and sidewalk neighborhoods.

Preserving sharp features in such noisy point clouds and preventing classes interference is not an easy task. Every splat in the generation phase with a normal $\hat{\mathbf{n}}_i$ will include a neighboring point $\mathbf{p}_i^k$ with a normal $\hat{\mathbf{n}}_i^k$ only if it passes the smoothness check $\hat{\mathbf{n}}_i \cdot \hat{\mathbf{n}}_i^k > \beta$ (we took $\beta = 0.6$). Once a point fails to pass this check, we stop growing the splat. Figure 3 illustrates the stopping cases.

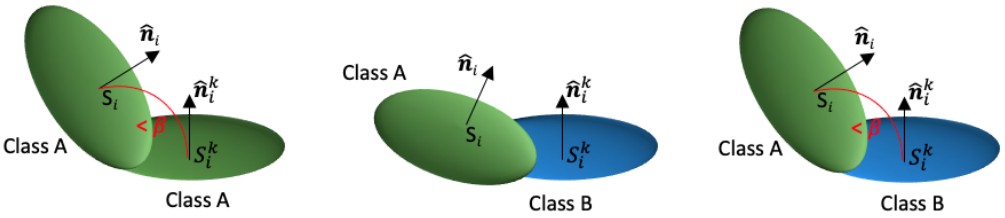

**Figure 3.** Illustrating the stopping cases to ensure the preservation of sharp features and avoid classes interference in the splats generation.

Performing automatic SS on point clouds using neural networks, or performing manual data annotation may not always be possible due to small scale data or time constraints. To lift this limitation from our proposed methods, we introduce a variant of AdaSplats that uses local geometric information to classify the points into the different surface groups and detail it below.

### 3.3. Adaptive Splatting Using Local Descriptors

We extend AdaSplats and provide a splats generation method that provides a better modeling quality than basic splatting in the absence of point-wise semantic labels. We call this variant AdaSplats-Decr (AdaSplats descriptors). When performing PCA on the neighborhood $\mathcal{N}_{\mathbf{p}_i}$ of a splat $\mathbf{p}_i$, we obtain the eigenvectors $(\hat{\mathbf{e}}_{i_1}, \hat{\mathbf{e}}_{i_2}, \hat{\mathbf{e}}_{i_3})$ and their corresponding eigenvalues $(\lambda_{i_1}, \lambda_{i_2}, \lambda_{i_3})$, such that $\lambda_{i_1} \geq \lambda_{i_2} \geq \lambda_{i_3}$, describing the spread of the data points in the local neighborhood, where the normal $\hat{\mathbf{n}}_i = \hat{\mathbf{e}}_{i_3}$ describes the direction of the lowest spread of the data.

Using the eigenvalues describing the spread of the three axes, the local neighborhood can be classified as linear, planar, or spherical by computing local descriptors [65] that reveal if the data are locally spread along one, two, or three dimensions, respectively:

$$Linearity = \frac{\lambda_{i_1} - \lambda_{i_2}}{\lambda_{i_1}}, \quad Planarity = \frac{\lambda_{i_2} - \lambda_{i_3}}{\lambda_{i_1}}, \quad Sphericity = \frac{\lambda_{i_3}}{\lambda_{i_1}} \tag{7}$$

The descriptor that best describes the local neighborhood is chosen by taking the largest value among the three. Using the computed descriptors, we divide the points in the point cloud into three main groups:

- Ground and surface using the planarity descriptor;
- Linear using the linearity descriptor;
- Non-surface using the sphericity descriptor.

We use the same values for the parameters $\mathcal{K}$, $\bar{\mathcal{R}}$, and $\bar{\mathcal{E}}$ on the linear and non-surface groups and find the best trade-off between a hole-free approximation and modeling accuracy for the Ground and Surface groups using $2\mathcal{K} = 80$, $2\bar{\mathcal{R}}$, and $2\bar{\mathcal{E}}$.

### 3.4. Splat-Based Resampling and Denoising

Point clouds collected using MMSs are subject to the presence of multilayered and noisy surfaces, which is due to sensor noise and localization errors of the MMS. As a preprocessing step before splats generation, we perform local denoising. Using the neighborhood $\mathcal{N}_{\mathbf{p}_i}$ of a given point $\mathbf{p}_i$, we compute the mean unsigned point-to-plane distance, compute the standard deviation $\sigma_i$, and remove a point $\mathbf{p}_i^k$ if its unsigned point-to-plane distance $|\epsilon_i^k|$ is greater than $3\sigma_i$.

Such point clouds also have highly varying densities, which are proportional to the distance between the sensor and the scanned surface. A high level of local anisotropy also dominates the point clouds, which is caused by the physical model of the LiDAR (sweeps of lasers) and can be observed from the high density of points along the sweep lines of the LiDAR and sparser in other directions. To reduce local anisotropy, we resample the point cloud based on the approximation of the surface from a first splats generation. After resampling, we restart the whole process of adaptive splats generation on the new point cloud.

First, we generate splats from point cloud $\mathcal{P}$ with our adaptive variant. To obtain prior information on the acceptable local density throughout the splat surface, we compute the average splats density $\bar{\delta}$ as the average number of splats in a spherical neighborhood of radius $\bar{\mathcal{R}}$, excluding splats belonging to the non-surface group. For a splat $S_i$, we start the resampling process whenever $\delta_i < \bar{\delta}$. We select the farthest splat $S_j$ in the neighborhood of radius $\bar{\mathcal{R}}$. We verify whether both splats belong to the same semantic class, or semantic group in the case of AdaSplats-Descr, and if they pass the smoothness check $\hat{\mathbf{n}}_{S_i} \cdot \hat{\mathbf{n}}_{S_j} > \beta$. If both checks are passed, we interpolate a new point that lies at the center of the segment connecting the splats' centers. If one of the checks fails, we iterate through the neighboring splats in descending order of distance to splat $S_i$ and re-check both smoothness and semantic class equality. We repeat the same procedure until the desired local density is achieved. Since we need both the LiDAR sensor position to reorient the normals and the semantic class for the splats generation, we assign to the new point the semantic class, or surface group for AdaSplats-Descr, and the LiDAR position of $p_i$ used to build splat $S_i$.

After the resampling step, adding the new points to the original point cloud $\mathcal{P}$, we get a more uniformly distributed point cloud $\mathcal{P}'$, which we use to restart our adaptive splats generation; this is able to fill small holes present before in the splat model. The smoothness and semantic class equality checks ensure that we do not smooth out sharp features.

## 4. Splat Ray Tracing

Simulating the sensors used by AVs, such as cameras and LiDARs, requires the implementation of an efficient ray–splat intersection algorithm. Very few approaches have worked on ray tracing splats to achieve high rendering quality. Ref. [26] was implemented to work only on CPUs, which makes it impossible to achieve real-time rendering. In this section, we introduce a ray–splats intersection method that leverages OptiX [64] to accelerate the process, achieving faster than real-time sensor simulation. OptiX is a ray-tracing engine introduced by NVIDIA that makes use of the GPU architecture to parallelize the ray-casting process and implements a BVH accelerating structure to accelerate the ray–

primitive intersection. Using our ray–splat intersection method, we can easily adapt our pipeline to include the simulation of other sensors, such as RADAR.

### 4.1. Ray–Splat Intersection

A splat is defined by its center $\mathbf{c}_{S_i}$, a normal vector $\hat{\mathbf{n}}_{S_i}$, and a radius $r_{S_i}$. To intersect the splat, we first need to intersect the plane in which the splat lies. To this end, we define the plane using $\mathbf{c}_{S_i}$ and $\hat{\mathbf{n}}_{S_i}$. A ray is defined by its origin $\mathbf{r}_0$ and a unit direction vector $\hat{\mathbf{r}}_{dir}$. The intersection point $\mathbf{p}_{int}$, in the world reference, along a ray can be found at position $t \in \mathbb{R}^+$ along $\hat{\mathbf{r}}_{dir}$:

$$\mathbf{p}_{int} = \mathbf{r}_0 + \hat{\mathbf{r}}_{dir} * t \tag{8}$$

A vector $\mathbf{v}_{i_k}$ can be computed from any point $\mathbf{p}_{i_k}$ lying on the plane with $\mathbf{v}_{i_k} = \mathbf{p}_{i_k} - \mathbf{c}_{S_i}$. This vector lies on the plane, so it is orthogonal to the normal vector, and this can be checked by taking the dot product $\mathbf{v}_{i_k} \cdot \hat{\mathbf{n}}_{S_i} = 0$. We cast a ray from the camera origin and compute $\mathbf{p}_{int}$ at a given $t$ and then report an intersection using the following check:

$$intersected = \begin{cases} 1 & if \quad (\mathbf{p}_{int} - \mathbf{c}_{S_i}) \cdot \hat{\mathbf{n}}_{S_i} = 0 \\ 0 & otherwise \end{cases} \tag{9}$$

solving for $t$

$$t * \hat{\mathbf{r}}_{dir} \cdot \hat{\mathbf{n}}_{S_i} + (\mathbf{r}_o - \mathbf{c}_{S_i}) \cdot \hat{\mathbf{n}}_{S_i} = 0 \tag{10}$$

with

$$t = \frac{(\mathbf{c}_{S_i} - \mathbf{r}_0) \cdot \hat{\mathbf{n}}_{S_i}}{\hat{\mathbf{r}}_{dir} \cdot \hat{\mathbf{n}}_{S_i}} \tag{11}$$

If a ray–plane intersection is reported, we check if the point of intersection $\mathbf{p}_{int}$ lies inside the radius of the splat:

$$\mathbf{v}_{int} \cdot \mathbf{v}_{int} < r_{S_i}^2 \quad \text{with} : \mathbf{v}_{int} = \mathbf{p}_{int} - \mathbf{c}_{S_i} \tag{12}$$

### 4.2. OptiX

The intersection algorithm explained above is implemented in CUDA to cast rays in parallel. On top of the parallel ray casting, OptiX provides the BVH acceleration structure that is used to accelerate the ray–primitive intersection. More specifically, the BVH is created on the host side (CPU) using the splats centers, where each splat is encapsulated in an (AABB) used to build the acceleration structure, with a side length equal to the diameter of the splat. The AABB serves as a first ray-primitive encounter, since when a ray traverses the BVH, it keeps traversing it until an AABB is intersected, which invokes the custom intersection program (e.g., ray–splat intersection). Once the BVH is created, the ray generation program can be invoked on the device (GPU) side, which generates the rays and calls the BVH traversal program. When the ray–primitive intersection program reports the intersection, the closest hit program is invoked to return the point of intersection and/or perform the custom shading and return the final pixel color. Otherwise, the miss program is invoked to return the background color and/or no point. The algorithm is illustrated in Figure 4.

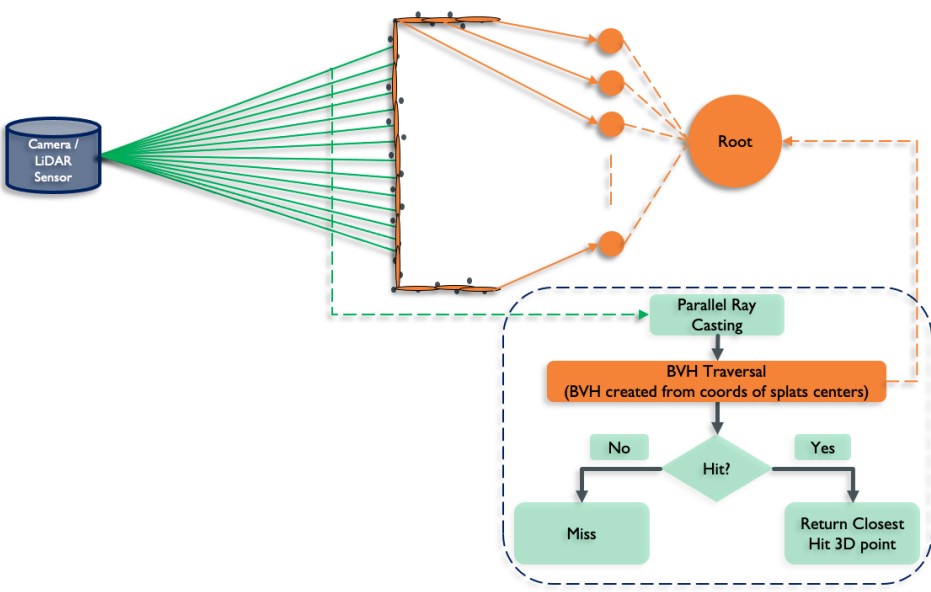

**Figure 4.** Parallel ray casting and accelerated ray–splat intersection are achieved using OptiX. The BVH is created from the splats primitives; then, the rays are cast in parallel on the device. Each ray traverses the BVH, and an intersection is reported back if a hit is found. Otherwise, the intersection for the specific ray is ignored.

## 5. LiDAR Simulation

Previous approaches [2,3] have not tackled the real-time aspect of LiDAR simulation. We focus our work on accelerating the sensor simulation and achieving real-time LiDAR simulation. To do this, we use the splat ray tracing method we introduced. We implement the firing sequence of multiple LiDAR sensor models, launch the rays in the splatted environment, and return the ray–splat intersection points, leveraging the high parallelization capability of the GPU (see Figure 5).

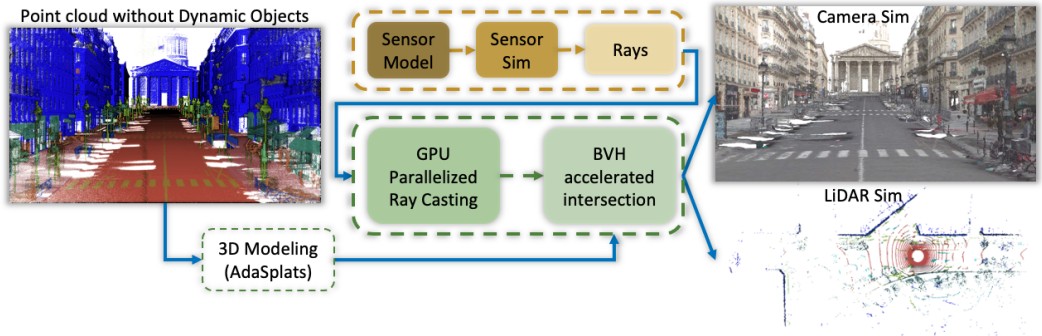

**Figure 5.** Our pipeline is split into different modules. The first generates accurate 3D modeling of the static environment using our adaptive splatting method. The second module takes the sensor model as input and simulates the sensor, including but not limited to camera and LiDAR, and generates the corresponding rays. In the third module, the rays are cast in parallel using the GPU architecture, then a bounding volume hierarchy (BVH) structure containing the generated splats is traversed, and ray–splat intersection point or color information is reported to generate the LiDAR or camera output, respectively.

### 5.1. Firing Sequence Simulation

Using our pipeline, we can simulate a variety of LiDAR models. In this work, we focus on simulating two LiDAR models, namely the Velodyne HDL-64 and the Velodyne HDL-32 LiDAR sensors. These LiDARs have an emitter and a receiver; they emit vertical equally spaced laser beams and compute the Time of Flight (ToF) of each beam to deduce

the distance to the object in the reference frame of the vehicle that reflected the light pulse. An azimuth revolution is a full 360° azimuth turn. We place the simulated LiDAR in AV configuration (roll, pitch, and yaw angles = 0°) and approximate the origin of the sensor beams by making them equal to the sensor position. All data coming from one revolution (a revolution is a full 360° azimuth turn) correspond to one LiDAR scan.

### 5.1.1. Velodyne HDL-32

The Velodyne HDL-32 emits 32 vertical laser beams. It has an azimuth angular resolution of 0.2° at 10 Hz with a 360° horizontal field of view (FOV). The sensor has an elevation angular resolution of 1.33° and ranges between −30.67° and +10.67°, summing to a 41.34° vertical FOV. Moreover, each laser emits 1800 laser pulses per azimuth revolution, which sums to 57,600 laser pulses across the 32 lasers. Finally, this sensor operates at a frequency between 5 Hz and 20 Hz and has a range of 100 m.

### 5.1.2. Velodyne HDL-64

The Velodyne HDL-64 emits 64 vertical laser beams. It has an azimuth angular resolution of 0.16° at 10 Hz with a 360° horizontal FOV. The sensor has an elevation angular resolution of 0.419° and ranges between −24.8° and +2.0°, summing to a 26.8° vertical FOV. Moreover, each laser emits 2250 laser pulses per azimuth revolution, which sums to 144,000 laser pulses across the 64 lasers. Finally, this sensor operates at a frequency between 5 Hz and 15 Hz and has a range of 120 m.

### 5.1.3. Firing Sequence Rays Generation

When simulating the LiDAR in AV configuration, there are no initial roll, pitch, or yaw angles, which results in a 360° rotation around the vertical axis (z in our virtual environment). If an initial transformation is required (e.g., initial pitch on the y axis), we pitch the initial ray $\mathbf{r\hat{a}y}^1$ by an angle $\eta^1$:

$$\mathbf{r\bar{\hat{a}}y}^1 = R_y(\eta^1)\mathbf{r\hat{a}y}^1 \tag{13}$$

where $R_y(\alpha)$ is the rotation matrix around the $y$-axis of angle $\alpha$.

If the LiDAR has no initial adjustments (roll, pitch, and yaw angles are equal to 0°), the previous step is ignored, and $\mathbf{r\bar{\hat{a}}y}^1 = \mathbf{r\hat{a}y}^1$. We perform the elevation and azimuth rotations to generate the rays for the LiDAR frame at timestamp $ts$. We accumulate the angles using the angular resolution of the simulated LiDAR and transform $\mathbf{r\bar{\hat{a}}y}^1$ into the ray corresponding to the new angle (azimuth on z and elevation on y). We first apply the azimuth rotation, followed by $n$ elevation angles for the $n$ laser beams:

$$\mathbf{r\hat{a}y}_{ts}^{i,j} = R_z(\psi^i)R_y(\eta^j)\mathbf{r\bar{\hat{a}}y}^1 \tag{14}$$

where $R_z(\beta)$ is the rotation matrix around the $z$-axis of angle $\beta$. $\psi^i$ and $\eta^j$ are the azimuth angle i and elevation angle j, respectively. For the rays, we only need the direction of the vector, and its position is provided separately. Consequently, when launching the rays, we pass its origin (LiDAR position) and direction.

## 6. Experiments and Results

We designed experiments to test the accuracy of our AdaSplats approach and the ability of our ray tracing method to achieve real-time performance on rendering and LiDAR simulation.

Validating the correct modeling of the environment to simulate a LiDAR is a complex task. The only experiments done by LiDARsim [3] and AugmentedLiDAR [2] were comparisons of learning results of deep networks between their simulated 3D data and data simulated under a simplified synthetic environment from CARLA [1]. In this section, we detail the task protocol and how we compare the performance of the LiDAR simulation using our splatting technique (AdaSplats) and other surface representations. We split the

experiments and results according to the different datasets that we use. Moreover, we show that using deep learning for semantic segmentation achieves similar results to AdaSplats using ground truth semantic information.

### 6.1. Experiments

To be able to precisely compare different modeling techniques, we choose three different datasets: two acquired by different mobile LiDAR sensors and mounted in different configurations and one built from a terrestrial laser scanner (TLS) (see Figure 6). Moreover, the datasets used were acquired in different cities around the world and in different environments (e.g., urban and suburban). We generate the scenes from the different datasets, using the different surface representations, then we render the scenes and simulate the Velodyne HDL-64 LiDAR in AV configuration using the generated splats. Finally, we provide quantitative results to evaluate the rendering quality provided by the different surface representations and perform quantitative evaluation of the different methods using the point cloud resulting from the accumulation of the simulated LiDAR scans.

For all of the experiments, we use our ray–splat intersection method implemented with OptiX and the original ray–triangles intersection of OptiX for meshed models. All experiments were done using an Nvidia GeForce RTX2070 SUPER GPU.

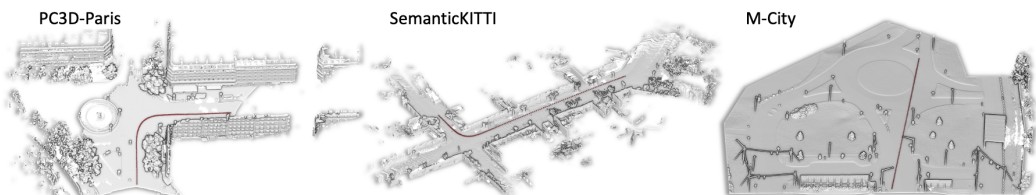

**Figure 6.** Point clouds used in the experiments (left to right:) PC3D-Paris, SemanticKITTI, and M-City. In red, we show the trajectory used for simulation.

### 6.1.1. Surface Representation

To demonstrate the accuracy of our approach, we chose to compare it to three other methods, namely Basic Splats, IMLS [8], and Screened Poisson [32]. Basic splats can be considered as a baseline representing the geometric precision of concurrent methods based on splatting, namely LiDARsim [3] and AugmentedLiDAR [2]. IMLS [8], and Screened Poisson [32] are two well-known and efficient meshing techniques.

We use the latest available code (version 13.72) for Screened Poisson surface reconstruction to mesh the point cloud, obtaining the finest mesh with: octree depth of 13, Neumann boundary constraints, samples per node as 2, and other parameters as default values. We also used SurfaceTrimmer to remove parts of the reconstructed surface that are generated in low-sampling-density regions with trim value as 10.0 and other parameters as default. For IMLS, we use a voxel size of 7 cm and fix the parameter $\sigma$ to two times the voxel size. We perform a sparse grid search with a truncation of three voxels, where we fill the IMLS SDF values only in voxels near the surface. Finally, we use the marching cubes algorithm [31] to extract the iso-surface.

### 6.1.2. Datasets

To test the performance and robustness of the different representations, we choose three different datasets: two acquired by mobile LiDARs, namely PC3D [19] and SemanticKITTI [18], and one built from a TLS (M-City).

#### Paris-Carla-3D

We use the PC3D-Paris part from the PC3D dataset [19], which is a dataset acquired in mapping configuration (LiDAR pitched at a 45° angle). The high diversity of complex objects (barriers, street lamps, traffic lights, vegetation, facades with balconies) present in this dataset allows us to compare the modeling capacities of the different techniques

in real situations. We remove dynamic objects using the semantic information; that is, we use ground truth semantic labels to remove the dynamic objects for all the methods excluding AdaSplats-KPConv. To remove dynamic objects for AdaSplats-KPConv, we use the corresponding predicted point-wise semantic classes. In a last step, we apply the different surface representations on the static background.

In the Results section, we first provide qualitative evaluation of the different $\mathcal{K}$-nn values to validate the choice of the parameters used for the different semantic groups. For the comparison between the different surface representation techniques, we use only two parts of PC3D-Paris, namely Soufflot-1 and Soufflot-2, containing 12.3 million points. Comparisons were made to determine the ability of the different methods to represent the geometry.

SemanticKITTI

Validating the accuracy of our modeling method requires testing it on other types of data. More specifically, we use a SemanticKITTI [18] dataset, which was acquired using a different sensor mounted in a different configuration with respect to PC3D (mapping), namely the Velodyne HDL-64 in AV configuration. The different sensor and mount result in a different scan pattern, which helps in validating if our method generalizes to other types of sensors or data. Moreover, the geometry in SemanticKITTI is different to that in PC3D-Paris, since it had been acquired in a different country and not in the heart of a large city. The same as in PC3D-Paris, we first remove dynamic objects using the semantic information and apply the different surface representations on the static background.

For the comparison between the different surface representation techniques, we use the first 150 scans from sequence 00 accumulated with a LiDAR SLAM [66], which results in a point cloud with 15.5 million points.

M-City

We take the testing even further and model a point cloud that was acquired using a TLS, namely M-City. This is an in-house dataset that was acquired in an AV testing site in Michigan using a TLS (Leica RTC360), at fixed points in a controlled environment without dynamic objects. The acquired point cloud was then used by 3D artists to perform manual surface reconstruction, which took 4 months. For this work, we use 25% of the original point cloud, which contains 17.5 million points. The interest in using M-City lies in the fact that it has a completely different scanning pattern, since the point cloud had been acquired at fixed points from a TLS.

For M-City, we do not have the sensor positions to re-orient the normals. Not having a correct normal orientation makes it difficult to properly reconstruct the surface using automatic meshing techniques (Poisson and IMLS). Having said that, for the comparison between the different surface representations, we compare the Basic Splats and AdaSplats approaches to the manually reconstructed scene mesh.

*6.2. New Trajectory Simulation*

To simulate the LiDAR sensor, we generate new trajectories. We shift the original trajectories provided by the sensor positions in PC3D and SemanticKITTI. As for M-City, since we do not have the sensor positions, and because the dataset had been acquired using a TLS, we perform the simulation on interpolated 3D points on a linear line segment.

For PC3D-Paris and SemanticKITTI, we take the original sensor positions provided with the datasets and offset them on the three axes (see Figure 6). More precisely, the offset for PC3D-Paris is $[1.0, 1.0, -1.0]^T$ along the $x$, $y$, and $z$ axes and $[1.0, 1.0, -0.5]^T$ for SemanticKITTI. For M-City, we generate a linear trajectory across the dataset.

Evaluation Metric for LiDAR Simulation

We measure the accuracy of the different models by computing the Cloud-to-Cloud (C2C) distance between the simulated point clouds (accumulation of all simulated LiDAR scans) and the original point clouds (used to model the environments):

$$C2C(\mathcal{P}_{sim}, \mathcal{P}_{ori}) = \frac{1}{|\mathcal{P}_{sim}|} \sum_{x \in \mathcal{P}_{sim}} \min_{y \in \mathcal{P}_{ori}} ||x - y||_2 \tag{15}$$

where $\mathcal{P}_{sim}$ and $\mathcal{P}_{ori}$ are the simulated and original point clouds, respectively, $x$ corresponds to a point in the simulated point cloud, and $y$ its nearest neighbor in the original point cloud. It should be noted that our C2C metric is not symmetric; we only calculate the distance from $\mathcal{P}_{sim}$ to $\mathcal{P}_{ori}$ because not all points in $\mathcal{P}_{ori}$ exist in $\mathcal{P}_{sim}$. For the closest point computation, we use the FLANN library implementation of a KDTree.

*6.3. Results*

6.3.1. Paris-Carla-3D

We start by validating the choice of parameters used for each semantic group. To do so, we generate the basic splats model on PC3D-Paris using different $\mathcal{K}$-nn, with $\mathcal{K} = 10$, 40, and 120, and compare the results to our AdaSplats method (see Figure 7). For a fair comparison with Basic Splats on the rendering and LiDAR simulation sides, we found that the best trade-off between geometric accuracy and a hole-free approximation is to use $\mathcal{K} = 40$. This results in holes on the surface, especially the ground. A larger $\mathcal{K}$ would result in very large splats, and a smaller $\mathcal{K}$ would result in many holes, which would not express well the local geometry.

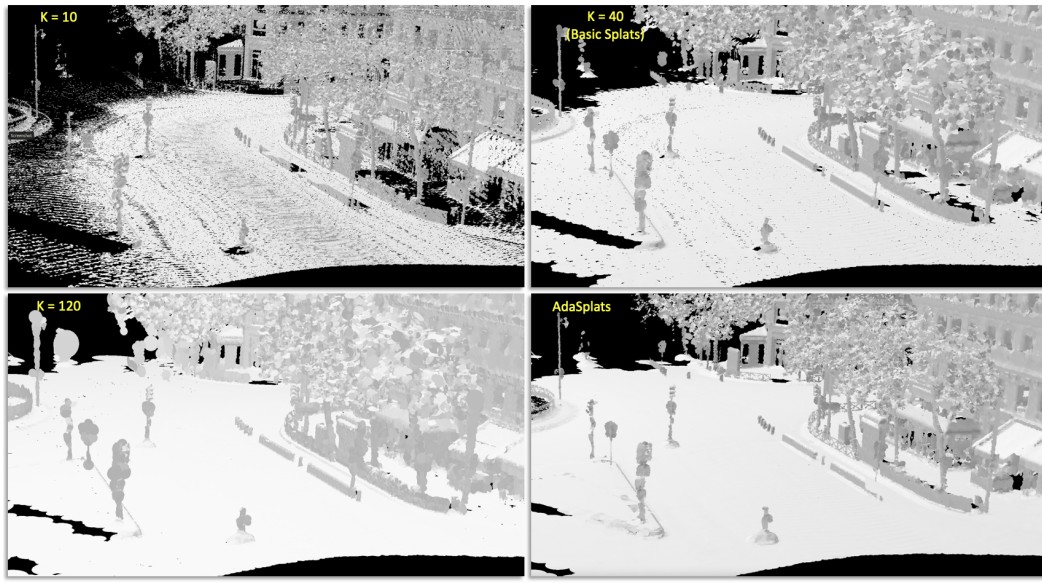

**Figure 7.** Rendering results on different choices of $\mathcal{K}$-nn on PC3D-Paris dataset. A small $\mathcal{K}$ (e.g., 10 or 20) results in holes on the surface and ground groups while also resulting in a better approximation on the non-surface and linear groups. A large $\mathcal{K}$ (40 to 120) results in a hole-free approximation of the surface and ground semantic groups while creating artifacts on small structures belonging to the linear and surface groups.

Figure 8 shows renderings of the different surface representation methods on PC3D-Paris dataset, where the last image (bottom right) shows the original point cloud. In these qualitative results, we show the modeling capabilities of the different surface representations. We can see that we obtain the best results with AdaSplats, especially on fine structures, shown in colored squares, thanks to the adaptiveness of our method.

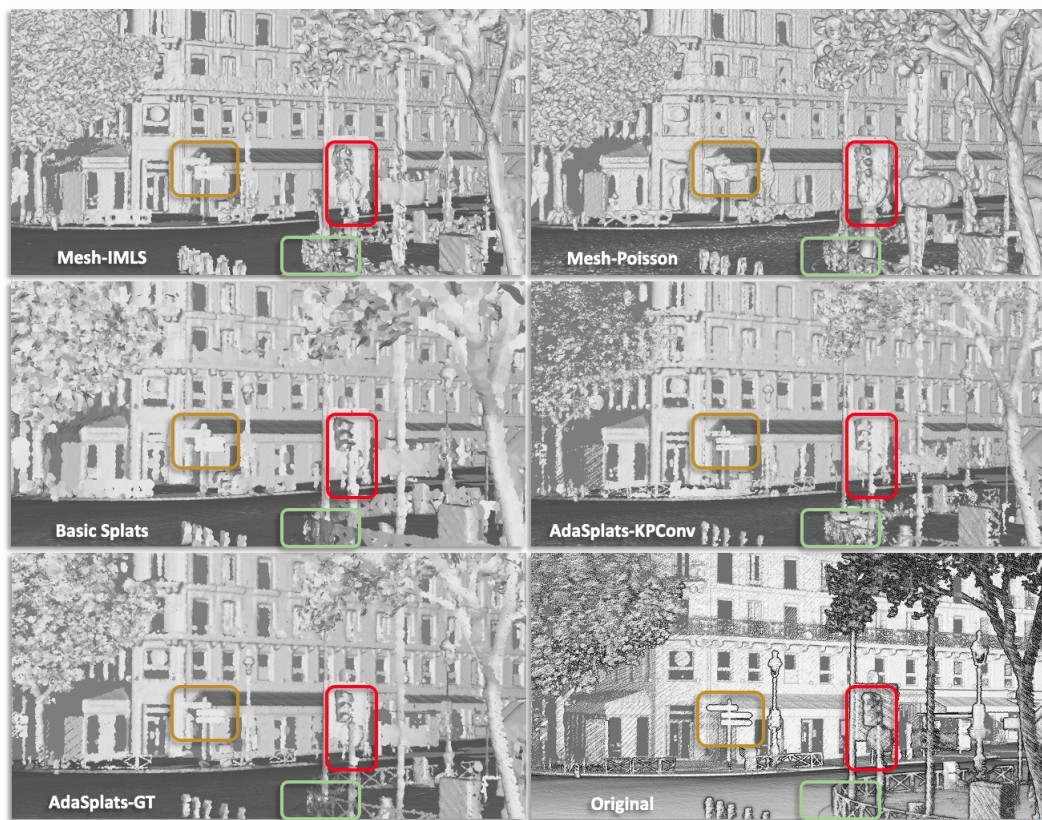

**Figure 8.** Rendering the different surface representations on PC3D-Paris. The top row shows the meshed scene using IMLS (left) and Poisson (right). The middle row shows the splatted scene using basic splats (left) and AdaSplats using KPConv semantics (right). The bottom row shows the splatted scene using AdaSplats-GT, which contains the ground truth point-wise semantic information (left) and the original point cloud (right). We show the ability of AdaSplats to recover a better geometry, especially on fine structures (in green, red and yellow boxes).

Figure 9 illustrates qualitative results comparing the accumulated point clouds from the LiDAR simulation with the different surface representations. The last image is the original point cloud used to model the environment. The other images are an accumulation of simulated scans (we show one simulated LiDAR scan in blue). We can see that AdaSplats results in higher-quality LiDAR data when compared to Basic Splats and other meshing techniques.

The simulation in meshed or basic splat environments does not perform well on thin objects containing few points, such as fences, poles, and traffic signs. Basic splatting techniques are not able to adapt to the local sparsity without semantic information. Screened Poisson [32] and IMLS [8] suffer from a performance drop on outdoor noisy LiDAR data, especially on thin objects. These surface reconstruction methods result in artifacts on open shapes; borders are dilated because these functions attempt to close the surface, as they are performing inside/outside classification. To limit this effect, we truncate the IMLS function at three voxels and perform surface trimming with Poisson. However, we can still see artifacts in Figures 8 and 9 (e.g., red, orange, and green areas).

Our method is also verified quantitatively on PC3D-Paris. We report the generation time of the different surface representations, rendering, and LiDAR simulation details of PC3D-Paris in Table 1. AdaSplats-KPConv includes KPConv for automatic SS (trained on the training set of PC3D dataset). On Paris-CARLA-3D, KPconv has an average mIoU of 52% over all classes and 68% IoU for the class "vehicles" (computed on test set Soufflot-0 and Souffot-3). AdaSplats-GT uses the ground truth semantic (manual annotation), and AdaSplats-Descr computes the local descriptors to arrange points into the three semantic

groups. We obtain the lowest number of primitives with our AdaSplats method. We observe that Basic Splats has the lowest generation time, and this is because there is no resampling, which results in generating the splatted environment only once. Moreover, the generation time of AdaSplats-KPConv includes the inference time of KPConv, which is 600 seconds for 10 million points. Adasplats-Descr achieves similar C2C distance to Basic Splats; however, it performs better on thin structures, as we see in the comparisons below. Moreover, it results in a lower number of primitives, which consequently accelerates the simulation and rendering time. The generation time of IMLS is very high, and we attribute this to our implementation, which could be improved. However, it would still result in a generation time higher than Screened Poisson.

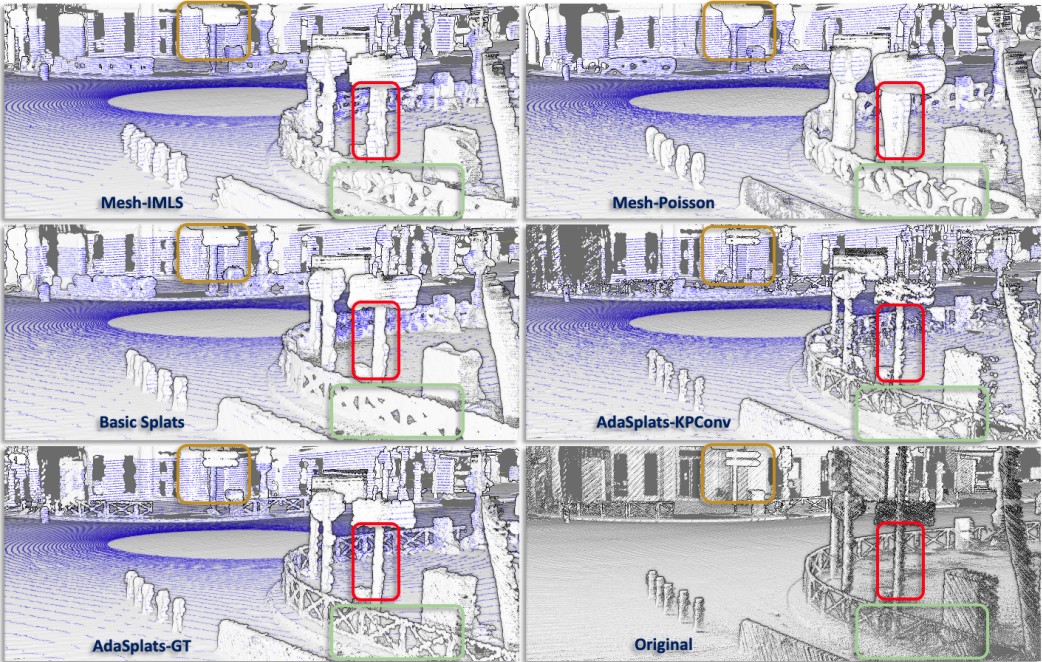

**Figure 9.** Comparison of simulated LiDAR data using different reconstruction and modeling methods on PC3D-Paris. The top row shows the simulation in meshed IMLS (left) and Poisson (right). The middle row shows the simulation with Basic Splats (left) and AdaSplats-KPConv (right). The bottom row shows the simulation with AdaSplats-GT (left) and original point cloud (right).

LiDAR sensors such as Velodyne HDL32 or HDL64 in default mode acquire one scan (full 360° azimuth turn) in around 100 ms (being at 10 Hz). The LiDAR simulation frequency (LiDAR Freq in Table 1) is the simulation time of one LiDAR scan. It includes generating the LiDAR rays at a given position, the host (CPU)-to-device (GPU) communication, ray-casting, primitives intersection, and reporting back the buffer containing points of intersection (device-to-host). Our ray–splat intersection is very fast, and we are able to simulate one scan in around 5 ms (203 Hz LiDAR Frequency for AdaSplats-KPconv in Table 1), being 20 times faster than real time. This is interesting for doing massive simulation. The rendering and LiDAR simulation frequency of meshed environments is higher, and this is expected because rendering pipelines are optimized to accelerate the ray–primitive intersection on polygonal meshes. Moreover, the ray–primitive intersection is hard-coded on GPU. However, we obtain a higher quality surface representation (Figures 8 and 9) and still achieve a rendering frequency that is faster than real time with our AdaSplats method. Furthermore, we obtain a lower LiDAR simulation frequency with respect to rendering frequency because it includes host–device (CPU-GPU) and device–host (GPU-CPU) communications for each scan, as we explain above, while the new frame position and rays generation for rendering is done on the device side.

**Table 1.** Results on PC3D-Paris. We report the time taken (Gen T) in seconds to generate the primitives (triangular mesh or splats), the number of generated primitives (Gen Prim) in millions (M), rendering frequency in Hz (Render Freq) with a resolution of 2560 × 1440 pixels, LiDAR simulation frequency (LiDAR Freq) of the Velodyne HDL-64, and the cloud-to-cloud (C2C) distance between the simulated and original point clouds.

| Model | Gen T (in s) | Gen Prim (#) | Render Freq (in Hz) | LiDAR Freq (in Hz) | C2C (in cm) |
|---|---|---|---|---|---|
| Mesh–Poisson | 797 | 5.20M | 1000 Hz | 232 Hz | 2.3 cm |
| Mesh–IMLS | 3216 | 6.32M | 920 Hz | 233 Hz | 2.0 cm |
| Basic Splats | 200 | 5.40M | 100 Hz | 135 Hz | 2.3 cm |
| AdaSplats-Descr | 344 | 3.90M | 160 Hz | 181 Hz | 2.3 cm |
| AdaSplats-KPConv | 1064 | 1.75M | 240 Hz | 203 Hz | 2.2 cm |
| AdaSplats-GT | 451 | 1.72M | 250 Hz | 205 Hz | **1.97 cm** |

We also report in Table 1 the C2C distance between the simulated and original point cloud (see Equation (15)). AdaSplats-Descr achieves similar overall accuracy, compared with Basic Splats. AdaSplats-KPConv improves the accuracy over Basic Splats, while AdaSplats-GT shows that, with improved semantics, the simulated data can be the closest to the original.

To see the effect of resampling on the final simulation, we remove the resampling step from the AdaSplats generation and report the results in Table 2. We observe that, without resampling, we obtain a higher number of geometric primitives, which affects the rendering frequency and results in a higher C2C distance. This demonstrates that, with our resampling technique, we are able to increase the accuracy of splats generation and lower the number of generated primitives thanks to the re-distribution of points.

Our resampling method does not result in a higher number of points; rather, it re-distributes the points and removes the excess in the form of noise and outliers. Moreover, it reduces the density throughout the whole point cloud. This re-distribution results in a better surface representation, which consequently reduces the number of overlapping splats in a given spherical neighborhood. Our resampling method preserves the hole-free approximation of the surface and sharp features thanks to the checks performed during the generation step.

**Table 2.** Results of the LiDAR simulation on the PC3D-Paris using AdaSplats with ground truth semantics without resampling (top row), compared to the simulation on the resampled model (bottom row). We report the time taken (Gen T) in seconds to generate the primitives, the number of generated primitives (Gen Prim) in millions (M), simulation frequency (Sim Freq) in Hz, and the Cloud-to-Cloud Distance (C2C) in cm between simulated and original point clouds.

| Model | Gen T (in s) | Gen Prim (#) | Sim Freq (in Hz) | C2C (in cm) |
|---|---|---|---|---|
| AdaSplats-GT no resampling | 169 | 2.84M | 180 Hz | 1.99 cm |
| AdaSplats-GT | 451 | 1.72M | 205 Hz | **1.97 cm** |

We notice that the point clouds contain a huge amount of points on the ground; this is the easiest class to model and has a higher effect on the computed distance. However, thin structures contain fewer points and are important for AV simulation. To measure the modeling of thin structures, we pick three classes from PC3D-Paris, compute the C2C distance on these classes, and report the results in Table 3.

We observe that AdaSplats (all variants) obtains much better results than IMLS, Poisson, or Basic Splats. AdaSplats-KPConv is able to achieve a C2C distance very close to

the model constructed with ground truth semantic information. We achieve a lower C2C distance on poles and traffic signs with KPConv due to misclassifications, leading to the generation of smaller splats.

We make an important observation, which is that we always obtain better quantitative and qualitative results independent from the source of the semantic classes. This proves that our method achieves better scene modeling, especially on fine structures, even if the semantic information is not perfect (see Tables 3 and 4).

**Table 3.** Cloud-to-Cloud distance (in cm) computed on PC3D-Paris for points that belong to classes of thin structures between the simulated and original point cloud. The AdaSplats methods include resampling.

| Model | Fences | Poles | Traffic Signs | Average |
|---|---|---|---|---|
| Mesh–Poisson | 5.9 | 6.1 | 6.7 | 6.2 |
| Mesh–IMLS | 4.6 | 3.5 | 2.9 | 3.7 |
| Basic Splats | 4.7 | 4.3 | 3.4 | 4.1 |
| AdaSplats-Descr | 4.1 | 3.7 | 2.9 | 3.2 |
| AdaSplats-KPConv | 5.5 | **2.1** | **1.1** | 2.9 |
| AdaSplats-GT | **2.4** | 2.3 | 1.8 | **2.2** |

We measure the contribution of resampling on thin structures we perform once more the semantic C2C distance on the AdaSplats model without resampling and report the results in Table 4. We observe a drop in performance, which can be seen from the higher distance we obtain between the simulated and the original point clouds on thin structures.

**Table 4.** Cloud-to-Cloud distance (in cm) computed on PC3D-Paris for points that belong to classes of thin structures between the simulated using AdaSplats without resampling and the original point cloud.

| Model | Fences | Poles | Traffic Signs | Average |
|---|---|---|---|---|
| AdaSplats-GT no resampling | 2.5 | 2.4 | **1.8** | 2.3 |
| AdaSplats-GT | **2.4** | **2.3** | **1.8** | **2.2** |

In Tables 2 and 4, we compare our AdaSplats method with and without resampling. By comparison, we obtain less primitives (1.72M) with resampling than without (2.84M). We also have a better repartition of splats on thin objects with resampling (2.2 cm) than without (2.3 cm).

AdaSplats is a method that uses, but does not require, perfect semantics, as can be seen from the simulation results inside the scene modeled using AdaSplats-Descr and AdaSplats-KPConv, which have errors. On the contrary, modeling methods that have specific models for semantic objects (e.g., a specific model for traffic lights) are highly dependent on the quality of the semantics and no longer work with the slightest error. Compared to mesh-based models using surface reconstruction, splats are independent surface elements whose parameters can be easily changed according to semantics, unlike methods based on SDFs, such as IMLS, or on an indicator function, like Poisson.

### 6.3.2. SemanticKITTI

Figures 10 and 11 show renderings and the simulated point clouds, respectively, using the different surface representation methods on SemanticKITTI. Looking at the results, we can see that using a point cloud acquired using a different sensor (Velodyne HDL-64) and mounted in a different configuration (AV configuration) does not change or reduce the quality of our AdaSplats method. We can observe that we are able to obtain the best quality

with AdaSplats, most noticeably on the traffic signs. We can also observe that we obtain the smoothest planar surface (the road) even when there are registration errors, such as in the sequence that we demonstrate in the figure. We can compensate small registration errors thanks to the resampling algorithm and our adaptive method, which takes a larger neighborhood into consideration for the generation of the splats on the ground.

We report the generation time of the different surface representations, rendering, and LiDAR simulation details of SemanticKITTI in Table 5. Viewing the results on SemanticKITTI, we can see that our modeling method is not limited to a specific LiDAR sensor or configuration. As a reminder, SemanticKITTI was acquired with a Velodyne HDL-64 in AV configuration, which is different from PC3D-Paris (acquired using HDL-32 in mapping configuration). The generation time of AdaSplats-KPConv includes the inference time of KPConv, which is 10 min for the first 150 frames of sequence 00. On SemanticKITTI, KPconv has an average mIoU of 59% over all classes and 94% IoU for the class "car" (computed on validation sequence 08).

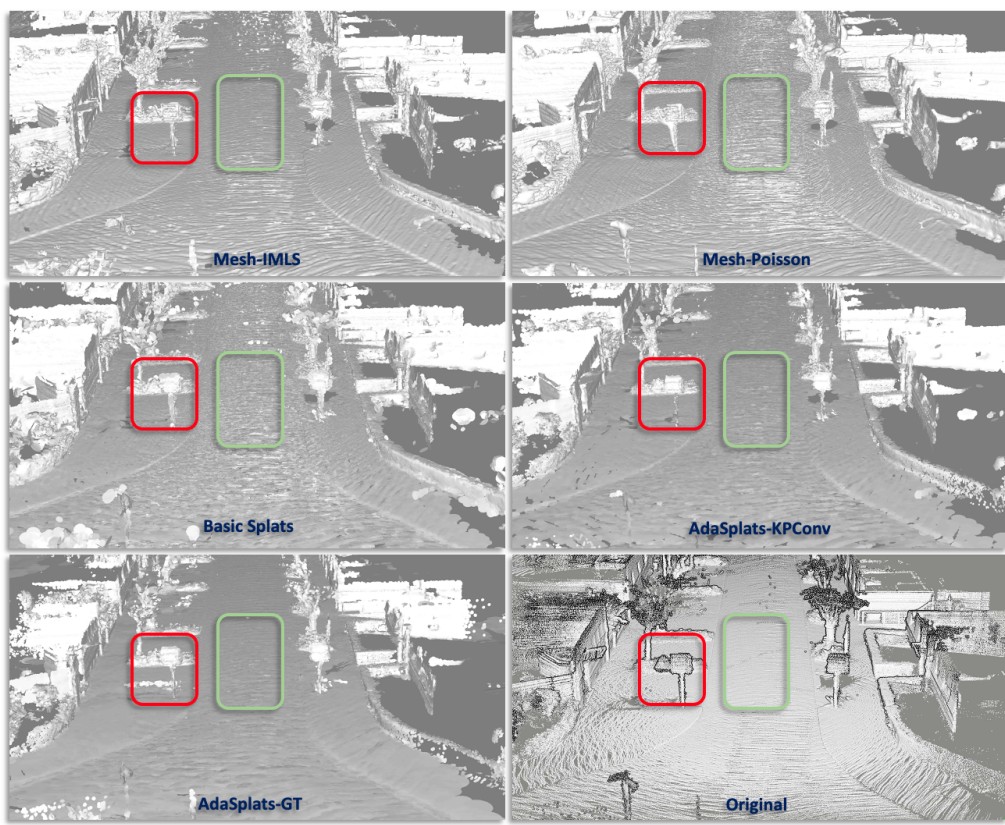

**Figure 10.** Rendering the different surface representations on SemanticKITTI. The top row shows the meshed scene using IMLS (**left**) and Poisson (**right**). The middle row shows the splatted scene using basic splats (**left**) and AdaSplats using KPConv semantics (**right**). The bottom row shows the splatted scene using AdaSplats-GT, which contains the ground truth point-wise semantic information (**left**) and the original point cloud (**right**).

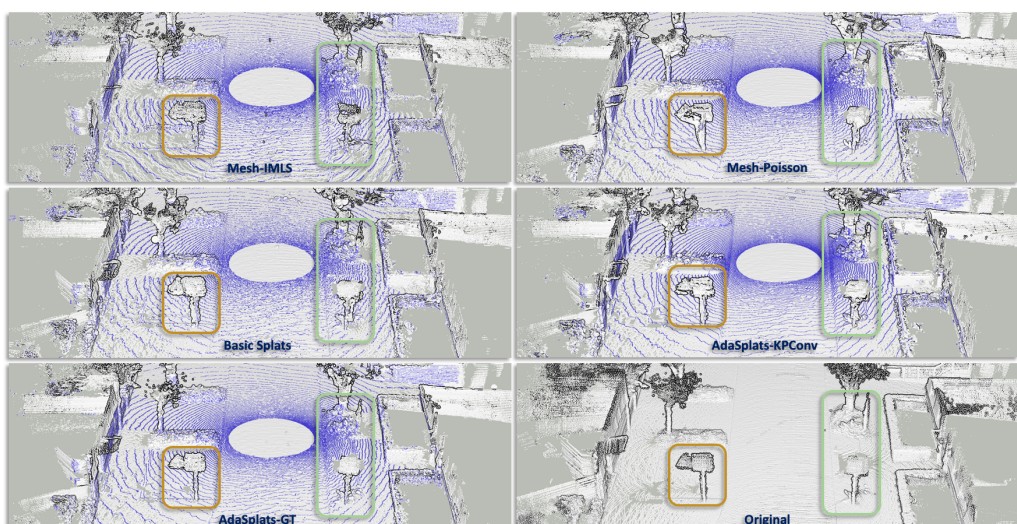

**Figure 11.** Comparison of simulated LiDAR data using different reconstruction and modeling methods on SemanticKITTI. The top row shows the simulation in meshed IMLS (**left**) and Poisson (**right**). The middle row shows the simulation with Basic Splats (**left**) and AdaSplats-KPConv (**right**). The bottom row shows the simulation with AdaSplats-GT (**left**) and original point cloud (**right**).

**Table 5.** Results on SemanticKITTI. We report the time taken (Gen T) in seconds to generate the primitives (triangular mesh or splats), the number of generated primitives (Gen Prim) in millions (M), rendering frequency in Hz (Render Freq) with a resolution of $2560 \times 1440$ pixels, LiDAR simulation frequency (LiDAR Freq) of the Velodyne HDL-64, and cloud-to-cloud (C2C) distance between the simulated and original point clouds.

| Model | Gen T (in s) | Gen Prim (#) | Render Freq (in Hz) | LiDAR Freq (in Hz) | C2C (in cm) |
|---|---|---|---|---|---|
| Mesh–Poisson | 796 | 5.97M | 1050 Hz | 229 Hz | 2.6 cm |
| Mesh–IMLS | 1380 | 7.05M | 1020 Hz | 222 Hz | 3.0 cm |
| Basic Splats | 185 | 7.77M | 170 Hz | 144 Hz | 2.6 cm |
| AdaSplats-Descr | 416 | 6.69M | 200 Hz | 156 Hz | 2.2 cm |
| AdaSplats-KPConv | 1166 | 6.11M | 220 Hz | 157 Hz | 2.2 cm |
| AdaSplats-GT | 544 | 4.56M | 240 Hz | 180 Hz | **2.0 cm** |

### 6.3.3. M-City

Figures 12 and 13 show renderings and the simulated point clouds, respectively, using the different surface representation methods on M-City. As a reminder, for M-City, we did not perform Poisson and IMLS surface reconstruction, since we do not have the position of the scanners to orient the normals. However, we use the manually reconstructed mesh to compare a manual reconstruction of the scene to Basic Splats and our AdaSplats method. Moreover, we cannot train KPConv on this small dataset; therefore, we do not include AdaSplats-KPConv in the comparisons. However, AdaSplats-Descr can be computed, and we can see that it is able to achieve lower C2C distance and higher simulation frequency compared to Basic Splats. Looking at the results, we can observe that our method (AdaSplats-GT) obtains a better surface representation, which can be clearly seen on the grass and vegetation that are hard to manually reconstruct due to the complexity of the geometry. When manually reconstructing complex geometry, 3D artists need to simplify the local geometry.

We report the generation time of the different surface representations, rendering, and LiDAR simulation details of M-City in Table 6. The same as PC3D-Paris and SemanticKITTI,

we are able to obtain the lowest number of primitives with our AdaSplats method. More-over, our automatic pipeline drastically reduces the generation time (more than 1 month for manual reconstruction against 8.5 min for AdaSplats), while obtaining a higher rendering quality. AdaSplats still provides accurate surface modeling capabilities, even without a correct normals orientation.

With M-City, we demonstrate that our pipeline can also be used on point clouds collected using a TLS, achieving LiDAR simulation results that are closer to reality than a manually reconstructed model. This is due to the modification of the local geometry done by 3D artists to simplify the reconstruction task (e.g., on the vegetation or some traffic signs).

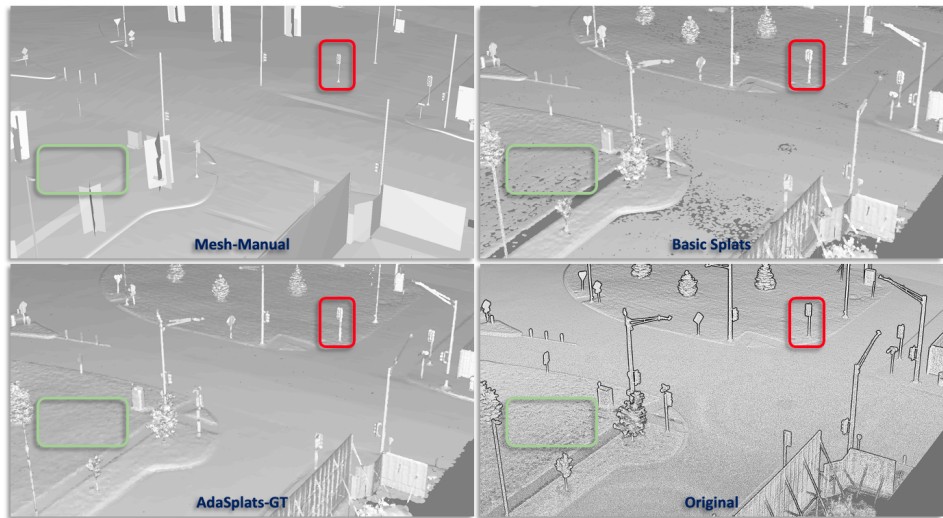

**Figure 12.** Rendering the different surface representations on M-City. The top row shows the manually meshed scene (**left**) and basic splats (**right**). The bottom row shows the results of rendering AdaSplats using GT semantics (**left**) and the original point cloud (**right**).

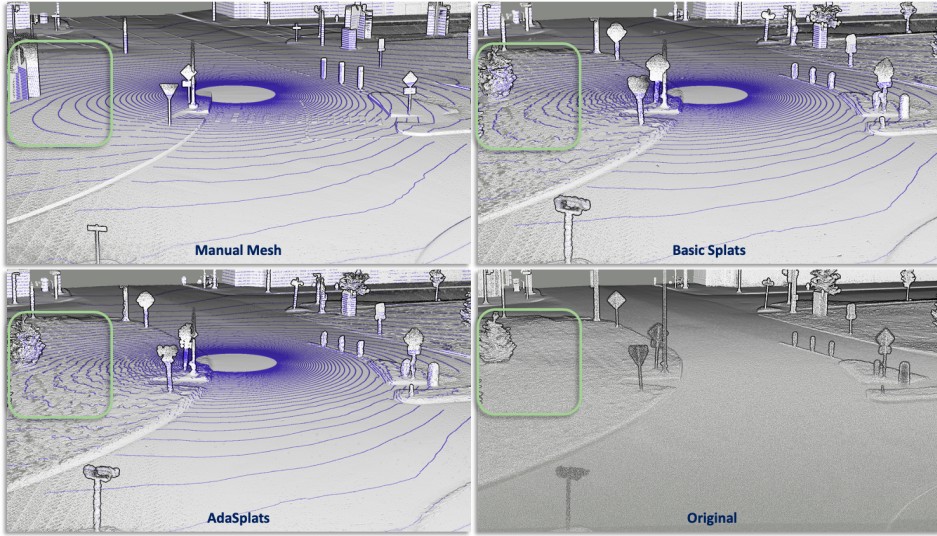

**Figure 13.** Comparison of simulated LiDAR data using different reconstruction and modeling methods on M-City. The top row shows the simulation in the manually meshed scene (**left**) and the modeled scene with Basic Splats (**right**). The bottom row shows the simulation with AdaSplats-GT (**left**) and original point cloud (**right**). Modeling vegetation is not an easy task and usually requires different ray–primitive intersection methods.

**Table 6.** Results on M-City. We report the time taken (Gen T) in seconds to generate the primitives (triangular mesh or splats), the number of generated primitives (Gen Prim) in millions (M), rendering frequency in Hz (Render Freq) with a resolution of 2560 × 1440 pixels, LiDAR simulation frequency (LiDAR Freq) of the Velodyne HDL-64, and cloud-to-cloud (C2C) distance between the simulated and original point clouds.

| Model | Gen T (in s) | Gen Prim (#) | Render Freq (in Hz) | LiDAR Freq (in Hz) | C2C (in cm) |
|---|---|---|---|---|---|
| Mesh–Manual | 1 month | 71.5K | 1930 Hz | 259 Hz | 7.0 cm |
| Basic Splats | 199 | 5.82M | 140 Hz | 110 Hz | 1.7 cm |
| AdaSplats-Descr | 480 | 3.92M | 290 Hz | 129 Hz | 1.6 cm |
| AdaSplats-GT | 513 | 3.01M | 440 Hz | 204 Hz | **1.5 cm** |

### 6.3.4. SimKITTI32

In the context of AVs, 3D SS methods [12–15] provide important information about the surroundings of the vehicles, increasing the level of scene understanding. Many manually labeled datasets are available [18,67]; however, not all datasets were acquired using the same sensor model or configuration. When SS networks are trained on a given dataset, they perform poorly when tested on datasets acquired using different LiDAR sensors, such as training on data acquired using a Velodyne HDL-64 and testing on datasets acquired using a Velodyne HDL-32. This is mainly due to the domain gap arising from the different sensor model, which affects the points density and scan pattern.

We introduce SimKITTI32, which is an automatically annotated dataset simulated using a Velodyne HDL-32 LiDAR sensor model in SemanticKITTI [18] sequence 08 (used in the validation procedure of 3D SS methods) that was acquired originally using a Velodyne HDL-64. SimKITTI32 is created with the aim to test the ability of SS methods to generalize to different sensor models. We use our AdaSplats method to model the full sequence 08 using the point-wise semantic labels provided with the original dataset. For the simulation of the Velodyne HDL-32 sensor, we use a slightly different LiDAR placement. More specifically, we use the original trajectory of the LiDAR sensor and offset its position by $-0.5$ m on the $z$-axis. This offset provides more scan lines on high elevations when simulating the Velodyne HDL-32, since it has a larger vertical field of view with respect to the Velodyne HDL-64.

First, we obtain the static scene by removing the dynamic objects from the dataset. Here, the dynamic objects refer to points belonging to the semantic classes of moving objects only while static objects, such as parked vehicles, are considered part of the static background. We extract frame-wise dynamic objects points and generate the splats on each frame separately. Due to the sparsity of points obtained from the moving objects, we generate one splat per point, with a fixed radius of 14 cm, which is equal to their average point-to-point distance.

We concatenate the splatted frames of the dynamic objects with the splatted static scene (using AdaSplats-GT) and simulate the Velodyne HDL-32 LiDAR. We make three improvements on the previous LiDAR simulation method.

First, we simulate the HDL-32 distance error computation with an additive white Gaussian noise with zero mean and $\sigma = 0.005$.

Second, instead of returning the first ray–splat intersection, we accumulate several intersections between the ray and the overlapping splats using a recursive call of the ray tracing function. More specifically, we define the depth ($\mathcal{D}$) of intersection ($\mathcal{D} = 5$ in our experiments), which defines how many overlapping splats we want to intersect along the same ray. The higher the depth, the more computations are required, which ultimately affects the simulation frequency, so we are left with a trade-off between precision and time complexity. Having defined the depth, we cast the rays from the sensor origin, traverse the BVH and return the data from the intersected splat, such as the center and the semantic class (the semantic class is saved with each splat during generation time). We

offset the intersection point by an $\epsilon$ (we use $10^{-4}$ in our implementation) to prevent self intersections, and cast a new ray; we then save the intersection data. We repeat these steps until the maximum depth is reached, or all splats of the same semantic class are intersected. Moreover, we put a threshold on the distance between two consecutive intersections (10 cm in our experiment), to prevent the accumulation of intersections belonging to different objects. If the number of overlapping splats is less than $\mathcal{D}$, they do not belong to the same semantic class, or the distance threshold is exceeded, we reset $\mathcal{D}$ to the maximum number of intersections. In a final step, we compute a weighted average of the intersection point, taking into consideration the depth of the intersection:

$$\mathbf{P}_{int} = \frac{\sum_{i=1}^{\mathcal{D}} \beta_i \mathbf{P}_i}{\sum_{i=1}^{\mathcal{D}} \beta_i} \tag{16}$$

where $\mathbf{P}_{int}$ is the final intersection point, $\mathbf{P}_i$ is the intersection point at depth i, and $\beta_i$ is a Gaussian kernel used to weight the contribution of each intersection to the final intersection point along the ray direction using the depth information:

$$\beta_i = e^{-|d_i - \frac{\mathcal{D}}{2}|/\frac{\mathcal{D}}{2}} \tag{17}$$

where $d_i$ is the current ray–splat intersection depth and ($\mathcal{D}$) the maximum depth of intersection.

Finally, when we concatenate all of the separate dynamic objects frames with the static background, we obtain trails of splats representing the displaced dynamic objects through time. If we simulate the LiDAR sensor directly on the concatenated splatted environment, we also obtain a trail of points. We make use of the semantic labels that we associate with the splat during generation time and check if the intersected splat belongs to a moving object. In that case, we invoke the any-hit program of OptiX to accumulate all the intersections along the ray. At generation time, we also assign the splats that belong to a moving object the corresponding frame number as an attribute, which we use at intersection time inside OptiX. If the number of the simulated frames matches the frame number of the splat intersected inside any-hit, we report only this intersection back. This ensures that we only intersect the splats belonging to the frame currently being simulated.

Figure 14 shows a frame from the SemanticKITTI dataset sequence 08, the same frame simulated with an HDL-64 LiDAR model at the same position without any offset, and the simulated frame using an HDL-32 LiDAR model shifted by $-0.5$ m on the $z$-axis. We can see that, with our implementation, we are able to accurately simulate the data with a different sensor model.

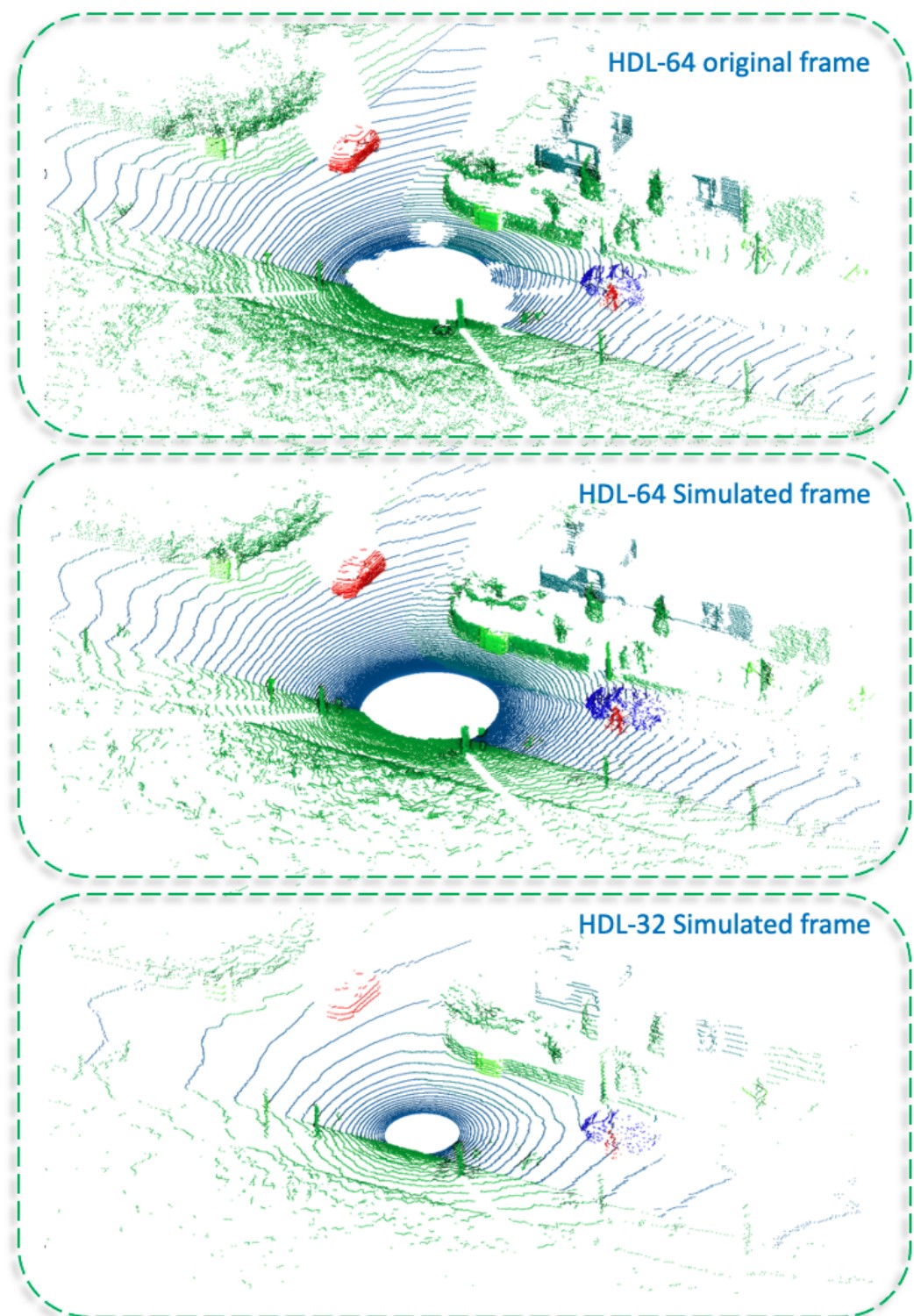

**Figure 14.** Showing an original frame from the SemanticKITTI [18] sequence 08 dataset with dynamic objects (**top**). The simulated HDL-64 LiDAR at the same position with dynamic objects (**middle**). The simulated HDL-32 LiDAR translated by $-0.5$ m on the *z*-axis (**bottom**).

## 7. Conclusions

In this work, we presented a simulation pipeline (AdaSplats) that leverages real-world data in the form of point clouds collected from mobile LiDARs, or fixed laser scanners. Our algorithm introduces the usage of semantic information to refine the generation of splats. Moreover, we present a novel resampling method to increase the uniformity of the points

distribution in a point cloud acquired in outdoor environments. The resampling method outputs a finer representation of the underlying surface in a noisy and non-uniformly distributed point cloud.

AdaSplats is able to achieve a higher 3D modeling capability when compared to basic splatting and other surface reconstruction techniques. Furthermore, we tested a LiDAR simulator in the splatted scene that leverages the GPU architecture and accelerates the ray–splat intersection using OptiX [64], achieving real-time sensor simulation performance in the splatted scene. Finally, we introduce SimKITTI32, a simulated dataset with a Velodyne HDL-32 LiDAR sensor inside a scene acquired using a Velodyne HDL-64 LiDAR. SimKITTI32 can be used to test the ability of semantic segmentation methods to generalize to different sensor models.

Our pipeline creates a "simulable" representation of the world, making it possible to generate in real-time virtual point clouds from simulated LiDAR sensors. The same pipeline could be used to generate virtual images from simulated cameras by adding textures to splats (and taking advantage of neural renderers such as NeRFs), but we leave that for further work.

**Author Contributions:** Conceptualization, J.P.R. and J.-E.D.; Software, J.P.R.; Formal analysis, J.P.R.; Investigation, J.P.R. and J.-E.D.; Writing—original draft, J.P.R.; Writing—review & editing, J.-E.D. and F.G.; Supervision, F.G. and N.D.; Project administration, J.-E.D. and F.G. All authors have read and agreed to the published version of the manuscript.

**Funding:** This research was funded by ANSYS, Inc.

**Data Availability Statement:** Data produced as part of this work can be found at: https://npm3d. fr/simkitti32 (accessed on 5 December 2022).

**Conflicts of Interest:** The authors declare no conflict of interest. The funders had no role in the design of the study; in the collection, analyses, or interpretation of data; in the writing of the manuscript, or in the decision to publish the results.

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
