# Peer review of "AdaSplats: Adaptive Splatting of Point Clouds for Accurate 3D Modeling and Real-Time High-Fidelity LiDAR Simulation"

_remotesensing, doi:10.3390/rs14246262_

Round 1
Reviewer 1 Report
In this paper, the author has mainly completed two parts of work. One is to use an adaptive splats generation method to accurately model the underlying 3D geometry from the data acquired by mobile mapping systems. It is a good idea to use deep learning to perform semantic segmentation on the raw point cloud, and use the labels obtained from these segmentation results to improve the modeling effect. The other work is about LiDAR Simulation. These work can make the test of autonomous vehicle algorithm separate from the real world and be verified in the simulation environment, which greatly reduces the risk and high cost of testing in the real environment. The method of this paper is very complete and the logic is clear. After reading this paper, I put forward the following problems that can be improved:
1. For the different parameters in lines 348 to 351, whether their value selection will have a significant impact on the results, because the author used completely different parameters in the preprint versions uploaded on arXiv (there are two versions in March and September) .
2. In line 385: 3.3. Splat based Resampling and Denoising, I only see the description of Resampling. Where did the description of Denoising go?
3. There are some presentation errors in the manuscript, some of which are listed below:
(1) Line 47: A consistent normals orientation (normal orientation?)
(2) Note to Figure 1: Mobile Maping System (Mapping?)
(3) Lines 122 to 125: My personal suggestion is that using more short sentences will make the content easier for readers to understand. In my opinion, this sentence and other sentences in the manuscript contain too many clauses that affect the understanding.
(4) Line 179: A previous methods overcomes this challenge.(methods or method?)
(5) Line 223: neighoborhood points,(neighborhood?)
(6) Line 314 -315, is the use of "its" correct here?
(7) Line 380: Linear using the linearity decriptor,(descriptor?)
(8) Line 627, perform surface trimmering(surface trimming?)
(9) Notes to Table 2: compared to the the simulation(There are two 'the')
(10) Line 823, which insures that(ensures?)
(11) Line 837: The combination of the introduced methods result in real-time(results in?)
Please check the rest for potential typos
Reviewer 2 Report
The authors present an interesting approach for 3D modelling of point cloud and then use the same for LiDAR simulation. Although the ideas are interesting, it appears that the authors have attempted to introduce too many things in the paper and therefore, have made the paper more confusing which otherwise would have been a very interesting paper. Following are my suggestions to the authors:
11 Problem needs to be defined clearly and objectives of the paper needs to be mentioned in a concise manner. The authors should clearly explain what they are trying to achieve, why is it relevant and what is the key novelty in their proposed approach.
2. All the abbreviations must be defined on first use. For example, “AVs” is not defined (line 16, page 1).
3. The authors introduce splatting on the very first page but define “splats” on page 10 in section 4.1. It should be defined earlier in the paper and its use should be justified.
4. The authors use specific methods and algorithms such as KPConv, but do not justify the choice of their method. Many of the methods are not clearly explained. For example, the authors do not explain how they detect moving objects, with what accuracy are they detected and removed, and their affect on the generated 3D model.
5. The authors demonstrate their results on MMS/TLS based datasets. Can the same approach be extended to UAV datasets or datasets collected from other platforms (Say mobile robots)? What modifications are needed in their approach to suit the proposed approach to different platforms.
6. The authors claim a “faster than real-time LiDAR simulator”. What is its meaning and significance?
7. Some of the sentences are too long. For example, lines 162-166 on page 5. These sentences are difficult to read and comprehend. Please consider breaking such sentences into shorter meaningful sentences.
8. Section 2 on related works needs significant improvements. It appears there are several categories of works, all of which have been listed under one heading. There are works on surface reconstruction, modelling, rendering etc. The authors have merged many of them into a single title. This makes this section unappealing to the reader. Please consider re-organizing this section into meaningful subsections.
9. I recommend the authors to include relevant figures/ diagrams/flowcharts to explain the methods/algorithms.
10. Algorithms can be explained in a better manner, instead of simply typing them as paragraphs (such as section 3.1). Also, please use figures to explain the methods.
11. How are parameters mentioned in section 3.2 chosen?
12. Connectivity between subsections presented in section 3 needs improvement. In the current form, the reader is unable to understand why a certain sub-section is followed by another sub-section.
13. While generating the 3D model, how do authors propose to fill the gaps caused due to occlusion during data capture?
14. In section 5 (LiDAR simulation), the authors neglect the effect of laser divergence, surface properties (intensity model) and multiple returns in LiDAR sensors. Consequently, the effect of these on the estimated coordinates is also neglected. This makes the simulation too simplistic and unrealistic.
15. All notations used in equations must be clearly defined. For example, see equation 15. The notation is unclear/incomplete.
16. What does C2C distance indicate? Why did the authors decide not to use a Euclidean distance-based measure?
17. In the context of above question, how do authors establish the point correspondences?
18. How are the numbers presented under column C2C in table 1 different (1.97 vs 2.0, 2.2 vs 2.3 cm)?
19. The authors report a LiDAR simulation frequency for HDL-64 to be 180-200 Hz. What does this mean? These numbers are unrealistic.
20. In the context of LiDAR simulation, in what coordinate frame is the simulated LiDAR data generated? The details of the involved coordinate transformations are missing.
21. Photos in figure 11 are too small and hence, they are difficult to understand.
Reviewer 3 Report
The experimental study is good. However, Novelty needs to be properly highlighted with clarity / clear section. How the new approach is different from the existing methods/approaches?
A detailed flowchart shall be added to emphasize the additional parameters/steps in the splat/texture. Many places its too wordy and repetitive, reducing the quality of manuscript.
The conclusion shall be rewritten with focus on concluded points.
Round 2
Reviewer 2 Report
The authors have addressed almost all the queries.
Reviewer 3 Report
Good.
best wishes